# Compositional Aspects of Beverages Designed to Promote Hydration Before, During, and After Exercise: Concepts Revisited

**DOI:** 10.3390/nu16010017

**Published:** 2023-12-20

**Authors:** Íñigo M. Pérez-Castillo, Jennifer A. Williams, José López-Chicharro, Niko Mihic, Ricardo Rueda, Hakim Bouzamondo, Craig A. Horswill

**Affiliations:** 1Abbott Nutrition, R&D, 18004 Granada, Spain; ricardo.rueda@abbott.com; 2Abbott Nutrition, R&D, Columbus, OH 43219, USA; jennifer.williams@abbott.com; 3Real Madrid, Medical Services, 28055 Madrid, Spain; jlchicharro@ext.realmadrid.es (J.L.-C.); nmihic@realmadrid.es (N.M.); 4Abbott Nutrition, R&D, Chicago, IL 60064, USA; hakim.bouzamondo@abbott.com; 5Department of Kinesiology and Nutrition, University of Illinois at Chicago, Chicago, IL 60608, USA; horswill@uic.edu

**Keywords:** carbohydrate–electrolyte solutions, CES, hydration, rehydration, hyperhydration, athletes, exercise, sports drinks, oral rehydration solutions, ORS

## Abstract

Hypohydration can impair aerobic performance and deteriorate cognitive function during exercise. To minimize hypohydration, athletes are recommended to commence exercise at least euhydrated, ingest fluids containing sodium during long-duration and/or high-intensity exercise to prevent body mass loss over 2% and maintain elevated plasma osmolality, and rapidly restore and retain fluid and electrolyte homeostasis before a second exercise session. To achieve these goals, the compositions of the fluids consumed are key; however, it remains unclear what can be considered an optimal formulation for a hydration beverage in different settings. While carbohydrate–electrolyte solutions such as sports drinks have been extensively explored as a source of carbohydrates to meet fuel demands during intense and long-duration exercise, these formulas might not be ideal in situations where fluid and electrolyte balance is impaired, such as practicing exercise in the heat. Alternately, hypotonic compositions consisting of moderate to high levels of electrolytes (i.e., ≥45 mmol/L), mainly sodium, combined with low amounts of carbohydrates (i.e., <6%) might be useful to accelerate intestinal water absorption, maintain plasma volume and osmolality during exercise, and improve fluid retention during recovery. Future studies should compare hypotonic formulas and sports drinks in different exercise settings, evaluating different levels of sodium and/or other electrolytes, blends of carbohydrates, and novel ingredients for addressing hydration and rehydration before, during, and after exercise.

## 1. Introduction

Body water homeostasis is maintained as a result of the continuous efforts exerted by the human brain through a complex and dynamic neuroendocrine network [1]. When compensatory responses needed to preserve this homeostasis are minimal, a state of euhydration exists [1]. Deviations from this balance can lead to states of increased or reduced total body water defined as hyperhydration and hypohydration, respectively. Achieving euhydration, or subtle hyperhydration, is relevant to the athlete since even modest levels of hypohydration have been shown to impair athletic performance [2].

Total body water is a fluctuating volume that comprises 50–65% of body weight, the proportion of which strongly depends on aspects such as age, sex, and body composition, and is distributed to intracellular (≈67%) and extracellular compartments (≈33%), the latter including both interstitial fluid and intravascular fluid (plasma) [3]. All these volumes are not static but are part of a dynamic balance, a fact that complicates the assessment of hydration status. Thus, a relatively simple technique for monitoring body water in exercising individuals is measuring acute changes in whole body mass to estimate the volume of water lost (dehydration) and assess the degree of hypohydration. Other methods aimed at evaluating hydration status following exercise-induced sweat loss include assessing urine color, specific gravity or volume, urine or plasma osmolality, and thirst sensation (methods for monitoring hydration in athletes have been reviewed elsewhere [4]).

Importantly, athletes are generally recommended to avoid fluid deficits of more than 2% total body mass loss (BML) during exercise as per most updated position statements and guidelines [2,5]. Although strongly dependent on environmental factors, 2–3% BML may deteriorate cognitive function and compromise aerobic exercise performance, while decrements of 3–5% BML have been shown to affect technical skills, and anaerobic and high-intensity exercise performance [5]. Physiological mechanisms involved have not been completely delineated but hypohydration seems to impair endurance exercise capacity through the reduction of cardiac output as a result of lower circulating blood volume, and this may potentially compromise skill and cognitive performance through closely associated factors such as hyperthermia, fatigue, and subjective discomfort or perceived exertion [6]. In addition, hypohydration is known to aggravate physiological mechanisms associated with impaired aerobic performance in hot environments through synergistic actions, such as increased cardiovascular strain and elevated core temperature [7], and might also be associated with increased levels of catecholamines [8] and enhanced glycogenolysis [9], thus potentially contributing to fatigue mechanisms. To prevent potential performance impairments, athletes are recommended to follow hydration plans in order to optimize their pre-exercise hydration status and compensate for fluid loss during and after exercise, thus minimizing the extent of hypohydration. Accordingly, the volume of fluid intake before, during, and after exercise should be tailored to the needs pertaining to the exercise performed, the characteristics of the athlete, and environmental conditions. The main recommendations from current consensus documents on fluid intake volume plans are presented in Table 1.

It is important to note that not only volume and timing but also the composition of the hydration beverage can notably impact the prevention or management of dehydration [2]. While consuming plain water may be sufficient for restoring fluid balance in individuals engaging in physical activity lasting less than an hour, sweat rates induced by higher duration/intensity exercise can lead to electrolyte losses, particularly sodium and chloride, which must be replenished along with water [2]. Such losses are subject to important inter-individual variability [13] and are exacerbated by several factors, including clothing, environmental conditions such as heat and humidity, and athlete’s acclimatization [14,15]. It is crucial to provide electrolytes during strenuous endurance activities as plain water intake exceeding sweat rate is considered to be a primary contributor to hyponatremia [16], a potentially life-threatening condition defined by plasma sodium levels <135 mmol/L (although clinically relevant at <130 mmol/L) [17].

Other compositional aspects such as additional energy sources (i.e., carbohydrates) and the resulting energy density and osmolality of the fluid consumed are critical for effective fluid replenishment since they impact gastric emptying and mechanisms involved in intestinal water absorption [18,19,20]. Further, components such as macronutrients and electrolytes can attenuate diuresis, which is important to promote fluid retention and sustain plasma osmolality during post-exercise rehydration [21,22]. As a result, formulations containing carbohydrates and electrolytes (hereinafter carbohydrate–electrolyte solutions (CES)) are typically designed for simultaneously replacing water and electrolyte losses, accelerating and maintaining rehydration during and after exercise, and sustaining fuel demands and replenishing glycogen stores.

Although subject to extensive research for decades [23,24,25,26], it remains unclear what compositions of CES aimed to improve hydration in athletes can be considered optimal in some settings, with loose ranges (i.e., consuming beverages with 20–50 mmol/L of sodium before exercise [10]) and general statements (i.e., “sodium should be ingested during exercise when large sweat sodium losses occur” [5]) being frequently recommended. This is typically justified based on the impacts that environmental conditions, exercise type and intensity, and athlete characteristics have on CES compositional requirements. Nonetheless, some authors have attempted to isolate the hydration properties of different beverage compositions by assessing urine output relative to plain water over four hours in euhydrated male subjects at rest in thermoneutral conditions, the ratio for which was defined as the beverage hydration index (BHI) [27]. This measurement, analogous to the glycemic index, has been shown to be independent of body mass index (BMI) [28], sex [28], and training level [29], but may be impacted by age, with older individuals retaining net fluid balance longer with higher sodium-containing drinks compared to young adults [30]. Although this approach is useful to evaluate the water retention properties of different beverages in controlled conditions and a euhydrated state, it is of limited utility when deciding on hydration strategies in sports since it does not consider the impact of exercise and heat stress on renal function [31]. Attempts to objectively define the optimal composition of CES for optimal hydration in exercise can be difficult, yet new untested commercial formulations continuously emerge in the market, citing benefits related to hydration or athlete performance improvements [32]. Naturally, the timing of their consumption (before, during, and after exercise) and their composition are of great importance in providing a clear rationale for their effective use in hydration plans.

To date, several evidence analyses have endeavored to evaluate optimal compositions of CES for fluid replacement in exercise [23,25,26]. Some of these extensive reviews have focused on the role of carbohydrates in improving exercise performance [23,26], while those focused on the hydration properties of CES ingredients often lack some important recent advancements in the field [25]. Therefore, the aim of the present work is to revisit optimal ingredients and their concentrations for CES aimed to address hydration and dehydration before, during, and after exercise, and to examine the latest developments within the field [25].

## 2. Materials and Methods

For the present narrative review, a non-systematic comprehensive search was conducted by one reviewer in the electronic databases and search engines PubMed (Medline), Embase (both accessed through ProQuest Dialog^®^), and Google Scholar using Boolean Operators (AND, OR, NOT) in combination with relevant search terms to create search strings. Examples of search strings included: (“Oral Rehydration Solution”) OR (“Sports Drink”) OR (“Sports Beverage”) AND “Exercise”, and (“Sodium Loading”) OR “glycerol” AND “Hyperhydration”. An initial search was conducted in February 2023 and was updated periodically afterward. English-written peer-reviewed scientific articles and books published from inception to July 2023 were retrieved. Non-published thesis dissertations were not considered for inclusion. Citations of selected records were screened to identify further publications. All records were managed using the reference management software EndNote^®^ X7.

## 3. Compositional Aspects of Carbohydrate–Electrolyte Solutions

Both sports drinks and oral rehydration solutions (ORS) are considered CES since carbohydrates and electrolytes are key ingredients in their formulations [33,34,35]. Sports drinks (often labeled as “isotonics” regardless of their osmolality) typically consist of isotonic or hypertonic solutions containing higher carbohydrate and lower electrolyte levels than ORS and are formulated for different purposes such as accelerating rehydration and enhancing fluid absorption but also providing easily digested energy during exercise [33]. On the other hand, ORS, which are hypotonic solutions, were originally conceived for the management of diarrhea-induced dehydration [36], but recent research has explored their potential role in improving hydration outcomes in exercise (as elaborated in later sections) [37,38,39,40,41]. Importantly, although ORS formulations were standardized by the World Health Organization (WHO) to contain 75 mmol/L sodium (range 60–90 mmol/L) among different compositional factors [42], other expert organizations such as the European Society for Paediatric Gastroenterology Hepatology and Nutrition (ESPGHAN) and the American Academy of Pediatrics (AAP) allow lower sodium levels (45–60 mmol/L) for the management of mild to moderate dehydration due to acute diarrhea [43,44], which may be compatible with compositions recommended by expert institutions in the field of sports. For example, the American College of Sports Medicine (ACSM) has recommended ingestion of a drink with 20–50 mmol/L sodium or salt-containing foods along with water before exercise to help stimulate thirst and retain fluid; during exercise, ingestion of enough fluid and electrolytes to avoid excessive dehydration (>2% BML) but sufficient carbohydrate to sustain exercise intensity at approximately 30–60 g/h and, finally, post-exercise, it is advised to replace the electrolytes lost via sweat and to consume fluid volume at approximately 1.5 times BML [10]. The main compositional aspects of standardized ORS and commonly marketed sports drinks are presented in Table 2.

Although CES are highly utilized among athletes of different sports disciplines to enhance hydration and help to maintain energy demands [48,49,50,51], scarce attention to compositional differences of these solutions is paid in current guidelines with liberal recommendations (i.e., consuming beverages with 20–50 mmol/L of sodium before exercise [10]) being provided. In the following sections, the role of CES ingredients in maintaining or restoring fluid and electrolyte balance is reviewed.

### 3.1. Electrolytes

Sodium (Na^+^) is the main cation in extracellular fluid and thus plays a major role in regulating plasma osmolality. Along with chloride (Cl^−^), it is one of the two main electrolytes lost in sweat with concentrations varying between 20–80 mmol/L [52]. Sweat sodium losses can be substantial, escalating with both exercise duration and intensity [53,54]. In the 1960s, it was first observed that sodium actively co-transports with glucose in the intestine, creating an osmotic gradient that facilitates water absorption in the intestinal tract [55]. Also, well-known is the critical role that sodium plays through the activity of the sodium–potassium ATPase, an enzyme that actively transports sodium out of renal tubules, creating a concentration gradient that drives water reabsorption [56]. This process helps retain water by decreasing urine volume output [57]. Further, sodium is crucial for stimulating thirst, as decreases in extracellular osmolality driven by large water intake without proper sodium replacement can inhibit the drive to drink and compromise adequate hydration [58]. Also, moderate sodium levels of the consumed beverage enhance palatability, which promotes fluid intake and subsequently impacts fluid replacement, while higher levels (i.e., >60 mmol/L) might decrease it [59,60,61]. Lastly, water and electrolyte loss, mainly sodium, is the most commonly theorized cause of exercise-associated muscle cramps (EAMC), yet empirical evidence is needed to demonstrate causality [62]. Overall, sodium is considered the most critical electrolyte when formulating a CES due to its well-established role in fluid and electrolyte balance.

While sodium is the main cation in extracellular fluid, potassium (K^+^) represents the main cation in intracellular compartments [63]. Although potassium is also excreted through sweat, concentrations are notably lower than those of sodium (4–8 mmol/L) [52], and the effect of exercise intensity on associated losses has been shown to be minimal [53,54]. Regarding fluid balance, the role that potassium plays seems unclear. In addition to helping to recover potassium sweat loss, adding potassium to CES compositions has been suggested to enhance water retention to a similar extent as sodium, and is proposed to favor intracellular hydration [64,65]. However, these findings might be equivocal, as shown in ensuing research observing no effect of additional potassium on fluid retention when sodium content is standardized [66]. The hypothesized increase in intracellular hydration at the expense of extracellular fluid could explain the lack of effect of potassium on increasing plasma volume, yet it is challenging to verify [64,65]. Higher sodium chloride concentrations in CES can compromise palatability and have been hypothesized to increase urine potassium levels, thus the addition of potassium to these formulations might support potassium urinary loss, and increase water retention without excessive sodium content in CES [67]. In addition to sodium, potassium CES content can contribute to restoring electrolyte balance, which is thought to prevent or ameliorate EAMC, but confirming data are still lacking [68,69].

Sodium and potassium are usually conjugated with chloride when added to CES compositions due to chloride being the main anion in extracellular fluid, and having a high excretion rate in sweat (20–60 mmol/L) [52], mainly as sodium chloride [70]. Chloride has been proposed to be at least partially responsible for the observed effects of potassium chloride (KCl) on restoring fluid balance after exercise [71], and, through an amiloride-independent interaction with salt taste receptors, can induce a salty, metallic taste, which is linked to the disagreeable palatability of very high sodium chloride (NaCl) concentrations [72]. The partial replacement of chloride with different anions such as citrate and bicarbonate has been proposed to improve the palatability of these formulations [24], and various sodium–anion conjugates have been explored as buffering agents in hyperhydration strategies [73]. Other electrolytes critical for muscle function, such as calcium (Ca^+2^) and magnesium (Mg^+2^), are relatively conserved by the body during exercise and heat exposure and have been shown to contribute minimally to hydration either as endogenous ions or beverage additives [23,52]. Nonetheless, adding small quantities of these ions to CES might be beneficial for individuals supplemented with high doses of caffeine, which has been shown to increase urinary losses of Ca^+2^ and Mg^+2^ [74,75]. Lastly, while Mg^+2^ supplementation has been explored to ameliorate or prevent idiopathic cramps in the elderly or pregnancy-associated cramps, no well-conducted research evaluating its role in EAMC is available to date [76].

### 3.2. Carbohydrates

The addition of carbohydrates to CES is primarily aimed at maintaining a high carbohydrate oxidation rate, which is key to supporting endurance capacity and performance during exercise and replenishing glycogen stores during recovery from exercise [77]. In addition, carbohydrates also play important roles in fluid balance. As mentioned, the intestinal co-transport of glucose and sodium facilitates water absorption. The sodium–glucose cotransporter 1 (SGLT-1) is responsible for the co-transport of glucose and galactose with sodium in the small intestine, while fructose is passively transported through the glucose transporter 5 (GLUT-5), thus not requiring the presence of sodium [78]. All these processes lead to an osmotic gradient that facilitates water absorption in the small intestine. Combinations of different carbohydrates, such as glucose/maltodextrin or fructose, as well as the addition of sucrose, a disaccharide of glucose and fructose, may improve water uptake by preventing saturation of SGLT-1 [79]. However, high amounts of fructose alone or fructose-to-glucose ratios exceeding one are typically discouraged in CES as they increase lumen osmolality due to limited fructose absorption [80]. The lack of studies evaluating fructose-to-glucose ratios > 1 is made apparent by reviews exploring optimal fructose–glucose ratios for endurance performance [79].

In addition to the volume of ingested fluid, the main contributor to modulating gastric emptying is energy density, with higher caloric beverages leading to slower emptying times [81]. Gastric emptying is important in the context of fluid balance since slower emptying rates decrease urine output, but delay increases in plasma volume, and might produce gastric discomfort [82,83]. Since delayed gastric emptying slows the movement of water into the systemic circulation, it consequently leads to sustained plasma osmolality, thus reducing urine production, and maintaining thirst and net fluid balance longer during rehydration [71]. However, whether these effects are beneficial from a rehydration standpoint is debatable since a delayed increase in plasma volume reasonably compromises effective rehydration. The delay might be particularly undesirable when the rehydration/recovery period before the second workout or competition is short. Although isoenergetic amounts of different macronutrients, such as proteins, lipids, and carbohydrates are emptied from the stomach at a similar rate [20], carbohydrates represent the main energy source of CES, and they are therefore the main ingredient affecting gastric emptying rates [84]. Reference concentrations at which gastric motility is impacted are subject to debate. While it is generally accepted that concentrations <6% carbohydrate do not significantly delay gastric motility at rest or during exercise compared to water [85,86,87,88,89,90], some authors have suggested that more subtle carbohydrate concentrations might impact gastric motility [91,92,93,94], while others have reported a lack of effect with higher concentrations [95,96,97]. Differences might arise from methodological aspects of available studies, namely the gastric emptying assessment method, impact of exercise, use of different types of carbohydrates and associated osmolality, ingested volume, and timing or frequency of intake [98].

Alternatively, some authors have taken advantage of stable isotope methodologies to integrate measures of gastric emptying and intestinal absorption [99]. Assessing the enrichment of stable isotopes (i.e., deuterium oxide (D_2_O)) in body water pools permits estimating rates of absorption from different beverages [100]. Millard-Stafford et al. reported that two sports drinks formulated with 6% and 8% carbohydrate content, respectively, produced similar fluid uptake measures (assessed through D_2_O intake), and changes in plasma volume and osmolality, though studied beverages differed in the type of carbohydrates and other compositional aspects [101]. In a comprehensive study, Jeukendrup et al. evaluated plasma D_2_O enrichment following intake of beverages containing increasing carbohydrate (0%, 3%, 6%, or 9%) or sodium content (0 mmol/L, 20 mmol/L, 40 mmol/L, or 60 mmol/L), thus enabling more direct comparisons [102]. In line with the abovementioned studies on gastric emptying, authors observed that increasing carbohydrate levels above 6% led to a decrease in fluid delivery compared to water [102]. Although this information is valuable for assessing the unidirectional movement of fluid, it is noteworthy that not all studies have observed consistent changes in D_2_O enrichment with different amounts of carbohydrates [103], or with different amounts of sodium in the presence of carbohydrates [102,104]. This might be linked to some methodological aspects of the methods employed (i.e., D_2_O enrichment does not account for fluid movement to the intestinal lumen [105]).

Due to the hydrophilic nature of glycogen, replenishing muscle glycogen stores might contribute to promoting intracellular water uptake and retention, which represents one mechanism through which CES carbohydrate content is speculated to contribute to fluid balance during recovery from exercise [106]. In fact, 1 g of muscle glycogen is frequently stated to be bound to 3 g of water [107], yet conflicting data are available, which might result from changes in levels of different glycogen structures [108]. As concluded in a recent analysis by King et al., glycogen-associated water does not seem to be an independent osmotically inactive reservoir, which implies that starting exercise in a glycogen-replete state might be beneficial for intracellular water content and whole-body hydration [109]. Lastly, the insulin response to the carbohydrate content of CES might play a role in water retention and maintenance of plasma osmolality. In this sense, the acute anti-natriuretic effects of insulin mediated through the promotion of renal sodium reabsorption are well established [110]. Formulations containing slow-digesting carbohydrates, such as isomaltulose, might elicit a lower but more sustained insulin response that might help maintain plasma osmolality over time [111]. Overall, CES containing moderate or high quantities of carbohydrates (i.e., >6%) might help retain sodium and increase plasma osmolality at the expense of delayed gastric emptying and slower increases in plasma volume.

### 3.3. Osmolality

CES osmolality, which is determined by the beverage’s total solute composition, depends mainly on the amount and type of carbohydrates present and, to a lesser extent, the electrolyte content. Thus, the role of osmolality in hydration can be independently assessed by separately evaluating simple carbohydrates and carbohydrate polymers [18].

In addition to the volume and energy content of ingested fluid, osmolality might also have a modest effect on gastric emptying of CES solutions [19,96,112,113], which depends on the amount and type of carbohydrates contained [18]. Shi et al. evaluated the impact that 2%, 4%, 6%, and 8% glucose or sucrose solutions have on gastric emptying rates in euhydrated males at rest [98]. Authors observed that the 8% sucrose solution (251 mOsm/kg) emptied faster than the 8% glucose solution (470 mOsm/kg), which was proposed to denote a combined effect of carbohydrate type and solution osmolality on gastric emptying [98]. Authors concluded that formulations containing over 6% glucose and 350 mOsm/kg total solutes can impede gastric emptying, which might be partially derived from their impact on duodenal and proximal jejunal osmoreceptors [98]. Recently, Sutehall et al. reported that the addition of sodium alginate and pectin to a maltodextrin (10.6%) and fructose (7.4%) polymer led to accelerated gastric emptying compared to a non-encapsulated polymer and a glucose (10.6%) and fructose (7.4%) solution at rest [114]. Despite a notable difference in the osmolality of the tested highly hyperosmotic beverages (around 1400 vs. 730 mOsm/kg), the impact on the emptying rate was modest [114]. These results support the notion that ~350 mOsm/kg might act as a saturation threshold at which hypertonic solutions impair gastric emptying, while further increases have little impact [114]. However, previous work conducted by Vist and Maughan reported notably slower emptying rates with a glucose polymeric solution of 1300 mOsm/kg compared to an isoenergetic glucose monomeric solution of 237 mOsom/kg [18], which might refute this threshold.

The osmolality of ingested CES can directly impact luminal content osmolality in the small intestine (270–290 mmol/kg) [100]. Water is rapidly absorbed as a consequence of its very low osmolality (≈30 mOsm/kg) when it reaches the duodenum, in a process aimed at regaining isotonicity of the luminal content [100,115]. In contrast, hypertonic solutions may reverse water flux by drawing it into the intestinal lumen due to its high osmotic content and will remain in the intestine longer until isotonicity or equilibrium is regained [100]. Observations of improved intestinal water absorption of hypotonic fluids led to the reformulation of the original ORS developed by the WHO from hypertonic to hypotonic, which helps increase intestinal water uptake using compositional maneuvers that do not compromise sodium absorption [116,117]. On the other hand, water absorption rates can differ based on the segment of the intestine analyzed, as intestinal permeability decreases towards further sections of the small intestine with narrow differences in osmolality having notable impacts on water absorption [100]. When isotonic and hypotonic CES reach the jejunum, water absorption is facilitated through carbohydrate co-transport, which explains why plain water may be absorbed slower than these solutions in this segment [100]. Further, water from hypotonic solutions (200–260 mmol/kg) is absorbed notably faster than isotonic solutions in the jejunum due to differences in the osmolality of the luminal content, which further supports the role of these formulations in maximizing rapid intestinal water absorption [100]. Along with carbohydrate content and type, the osmolality of ingested fluids may be the most important factor for optimizing water absorption in the proximal small intestine, as demonstrated by an analysis of 30 publications exploring triple-lumen tube intubation–perfusion techniques [118]. Whether providing multiple carbohydrates can offset the low jejunal absorption of isotonic or hypertonic solutions is debatable and further research is warranted [118].

### 3.4. Other Potential Ingredients

Proteins are key for promoting muscle anabolism and have been proposed to augment glycogen synthesis [119,120]. As protein-based products are typically consumed to support recovery post-exercise, the impact that this macronutrient may have on hydration outcomes has been explored by some authors [21,121,122,123,124,125]. Enhanced water retention properties of protein-containing beverages are largely based on their ability to delay gastric emptying. Accordingly, milk has been labeled as an effective rehydration beverage with better fluid retention properties than commonly available hypertonic CES, mainly based on its protein and electrolyte content [126,127,128,129,130]. It is important to mention that different types of protein can impact gastric emptying rates differently. For example, milk casein (the most abundant protein in milk) clots in an acidic environment, which leads to slower emptying rates compared to whey protein [131,132]. Isolates of milk protein have been tested to improve the water retention potential of commercial CES by different authors [21,122,123]. Most recent findings exploring different volumes and environmental conditions suggest that the addition of whey protein isolate to CES compositions might not significantly increase or decrease their water retention properties [122,124]. This has led these authors to speculate that the impact of protein on rehydration is specific to (whole) milk protein [125,133]. Nonetheless, the extrapolation of the effects of milk on rehydration and sustaining hydration after exercise in a practical setting is limited due to the high volumes explored (i.e., 1.5–2 L), which are far above the daily recommended milk intakes for adults [134]. Issues related to the high prevalence of lactose intolerance worldwide [135] pose further challenges to the use of milk as a rehydration beverage. On the other hand, protein supplementation in conjunction with exercise has been suggested to increase plasma volume through augmenting circulating albumin levels, which might support fluid retention [136,137].

Amino acids are primarily absorbed in the proximal intestine through mechanisms involving sodium co-transport [138], thus promoting water uptake following an osmotic gradient. Specific amino acids might be responsible for additional mechanisms that could promote water uptake. For instance, the production of nitric oxide (NO) is catalyzed by nitric oxide synthase (NOS) enzymes, which degrade the amino acid L-arginine to L-citrulline in the presence of the cofactor tetrahydrobiopterin (BH4) thus releasing NO in the process [139]. Endogenous NO has been shown to promote water absorption in preclinical experiments possibly through enhanced intestinal vasodilation [140,141]. Other amino acids, such as L-glutamine, have been reported to enhance sodium and water absorption even more effectively than glucose [142]. The addition of L-arginine and L-glutamine has been tested with attempts at designing an improved ORS [141,142,143]. Due to the instability and limited solubility of glutamine, dipeptides such as l-alanyl-l-glutamine, with improved physical properties, have been explored in CES formulations. In fact, CES containing l-alanyl-l-glutamine provided to hypohydrated subjects (2.5% BML) was shown to improve performance in a subsequent exercise bout compared to no fluid replacement or replacement of water only [144]. The improvements were thought to be partially mediated by the effects of l-alanyl-l-glutamine on electrolyte and water intestinal uptake. However, no changes in plasma osmolality, volume, or hormonal response were observed between treatments [144]. Subsequent studies showed a limited effect of rehydrating with l-alanyl-l-glutamine in terms of fluid balance or exercise performance compared to water [145], while others have supported a role in improving cognitive performance during intermittent sport, albeit no data on hydration/rehydration outcomes were reported [146]. Regarding amino acid-based ORS, a recent study compared the efficacy of a glucose-based ORS, a glucose-free ORS, and an amino acid-based glucose-free ORS on post-exercise rehydration when consumed at a volume of 125% BML [39]. Notably, authors observed that both glucose-based and amino acid-based glucose-free formulas restored fluid balance to a similar extent and better than the glucose-free beverage [39]. Lastly, it is possible that amino acid-based beverages might deliver additional benefits such as improved maintenance of the gastrointestinal epithelium during exercise-induced heat stress [147]. The use of amino acid-based formulas in exercise settings is an emerging field of research and future studies using various amino acids or dipeptides alone or in combination with carbohydrates may be of interest for CES hydration properties.

Caffeine is one of the most well-established ergogenic compounds to enhance endurance capacity, sustain attention, and reduce perceived exertion [148,149], which makes it an attractive ingredient for CES to be consumed before exercise. However, caffeine has been typically considered as a diuretic compound [150,151,152], which might promote urine production before and during exercise. The diuretic effects of caffeine have been proposed to be mediated by the inhibition of phosphodiesterases in the proximal tubule, and antagonism of adenosine receptors, which in turn leads to inhibited renal sodium reabsorption [153]. However, several studies have reported a lack of effect of caffeine intake on acute fluid balance [154,155], particularly when combined with exercise [156,157]. Similarly, mouth rinsing with caffeine does not appear to have any impact on hydration during exercise [158]. A pooled analysis of 16 studies reporting data on the impact of caffeine intake on urine output concluded that the mild diuretic effects of caffeine observed in resting individuals are blunted with exercise, which was shown to be independent of dose and timing [153]. The lack of effect of caffeine on diuresis during exercise has been speculated to be a consequence of compensatory catecholamine activity [157,159]. Of note, women appear to be more susceptible to the diuretic effects of caffeine compared to men [153]. Lastly, since CES volume intake can vary greatly, the selection of caffeine doses for CES poses important challenges. Doses employed to achieve the ergogenic effects of caffeine in experimental settings (i.e., 3–6 mg/kg body weight) might be associated with undesirable effects, such as nausea, anxiety, accelerated heart rate, and insomnia, in situations where beverages must be consumed in high amounts to compensate elevated sweat replacement needs, which may outweigh performance benefits [149].

Alcohol is a well-established diuretic whose mechanism is thought to involve the inhibition of vasopressin secretion [160] and the inhibition of renal tubular reabsorption [161]. Nonetheless, exercise/heat-induced hypohydration has been observed to blunt the diuretic effects of alcohol to a certain extent [161]. These findings have justified ensuing studies exploring low alcohol beer (≈2.5% alcohol content) as a potential hydration beverage with high social acceptance that may drive consumption thus improving rehydration. However, quantities as low as 4% alcohol, such as those contained in an average beer [162], have been reported to notably delay rehydration after exercise [160,163,164] and impair motor and cognitive performance in a further exercise bout [164]. Although adding electrolytes to these beverages might partially compensate for the deleterious actions of alcohol on post-exercise rehydration [165], there does not appear to be any clear rationale or benefit for the addition of alcohol to CES formulations [166].

In Figure 1, a schematic summarizing the main effects of CES compositional aspects on promoting hydration-related outcomes in exercise is presented.

## 4. Impact of Exercise Timing on Carbohydrate–Electrolyte Solutions

Hydration plans pursue different goals depending on the timing when the fluid should be provided. In the same manner, a one-size-fits-all approach for CES compositions to optimize hydration is complicated since the rationale supporting the use of different formulations to improve hydration status will also depend on the different needs of the athlete before, during, and after exercise.

### 4.1. Before Exercise

Ideally, athletes are expected to commence training and competition in a euhydrated state; however, this is not always the case. For instance, a recent systematic review of 24 studies reported that the prevalence of hypohydration, defined as >1.020 g/mL urine specific gravity, in professional soccer players was 66.2%, thus indicating a clear need for better pre-exercise hydration practices [167]. Avoiding pre-exercise hypohydration is critical as observed in a meta-analysis reporting reductions of 2.4%, 2.4%, and 4.4% in aerobic exercise performance, peak oxygen volume (VO_2_ peak), and lactate threshold, respectively, in hypohydrated compared to euhydrated individuals (−3.6 ± 1.0% vs. 0.3 ± 0.3% BML (mean ± standard error)) [168]. While the importance of education to improve adherence to fluid volume recommendations is obvious, there are some scenarios where CES compositions can play significant roles in influencing behaviors that promote pre-competition hydration status.

A classic situation is represented by combat sports athletes who, among other techniques, intentionally dehydrate to attain specific body weights to meet weight class requirements [169]. To ensure that extreme weight-cutting methods are not put into practice, short times are allowed between weight measurement and competition [170]. Since unreasonably large volumes of fluid would need to be replaced in a very short period, the composition of the fluid may be of particular importance if only partial replacement can be achieved. In order to facilitate fast recovery from self-imposed dehydration after weight measurement and before the commencement of competition, the use of CES [170] and buffering agents such as glycerol and sodium citrate has been utilized [171,172]. A recent ACSM consensus statement recommended replacing ≈150% of the fluid deficit, and consuming sodium-containing beverages (50–60 mmol/L) when hypohydration was achieved purposely via sweating [173]. In these scenarios, evidence on CES designed for post-exercise rehydration (discussed in later sections) should be taken into account, and further studies on pre-exercise hydration for combat sports following weight-cutting strategies are needed since this practice is likely to continue as the norm [174].

Before endurance efforts, hyperhydration strategies have gained great interest over time. Through the consumption of either water, glycerol, or sodium, oral hyperhydration attempts to increase total body water volume, which is particularly useful before exercise in situations where pre-exercise euhydration alone is insufficient to compensate the exercise-induced challenge in fluid balance, or when opportunities for fluid intake will be scarce [175]. The effectiveness of water as a hyperhydration solution is limited, as excess water when consumed in isolation will be excreted through urine [176]. On the other hand, glycerol is an osmotically active solute that is rapidly absorbed after oral intake and promotes water retention [177,178]. Glycerol has been extensively studied as a hyperhydration solute over the years [179]. In fact, glycerol was considered a masking agent and banned from competitions by the World Anti-Doping Agency (WADA) beginning in 2010 [180], to be later unbanned in 2018 [73]. The lifting of the glycerol ban has led to recent interest in this solute to promote hyperhydration in endurance athletes [181]. Nonetheless, meta-analyses of previous studies have observed that glycerol doses of 1.02 g/kg body weight in addition to 38.4 mL/kg water lead to only 3.3% increases in plasma volume compared to water alone, which indicates a limited hyperhydration potential in this solute [182]. Additionally, some studies have reported minor symptoms of nausea, bloating, and light headache after glycerol pre-loading [183,184,185]. A further concern of both water- and glycerol-facilitated hyperhydration consists of their potential diluting effect on plasma sodium concentrations, though the effect seems insufficient to induce hyponatremia in most situations [186].

The most preferred approach to pre-exercise hyperhydration consists of the use of sodium loading strategies, which have been reported to increase pre-exercise plasma volume, improve maintenance of plasma volume during exercise, reduce urine output, and even enhance endurance performance [187,188,189,190]. Compared to glycerol hyperhydration, sodium loading has been shown to lead to higher fluid retention over time (0–3 h) at rest [191,192]. Sodium levels explored in these studies are far higher (≈3–4 g/L) than those typically contained in CES formulas and have been frequently provided as salt tablets to bypass palatability issues. Nonetheless, liquid solutions have been reported to have better fluid retention properties than tablet formulations [192]. Notably, adding 6% dextrin to 120 mmol/L or 180 mmol/L sodium solutions was recently reported to accelerate sodium hypervolemic response in euhydrated individuals under resting conditions; and, therefore, this combination was proposed as an effective strategy to optimize timing when approaching hyperhydration strategies through hastening of the hypervolemic response, along with building on the evidence and science for hyperhydration and CES formulations [193].

One important aspect to consider is the amount of sodium needed to facilitate the hypervolemic response prior to exercise. The aforementioned study conducted by Fuji et al. reported no differences in cumulative urine or plasma volume between 120 mmol/L or 180 mmol/L sodium concentrations, thus denoting that sodium levels higher than 120 mmol/L might not lead to additional benefits in terms of changes in plasma volume [193]. In a previous study, water and three solutions containing 60, 120, or 180 mmol/L sodium, respectively, were compared [194]. In trials separated by three days, eight euhydrated males ingested 16–17 mL/kg body weight (≈1 L) of one of the three formulations or water within 60 min, and were followed a total of 150 min under controlled environmental conditions [194]. As expected, the 120 and 180 mmol/L solutions were shown to be more effective than the 60 mmol/L solution or water at reducing cumulative urine volume and promoting net fluid balance [194]. However, gastrointestinal symptoms (diarrhea) were only reported in subjects allocated to the 120 (one subject) and 180 mmol/L treatments (six subjects) [194]. Interestingly, a reduction in urine output and an increase in net fluid balance at 120–150 min was observed in individuals drinking the 60 mmol/L sodium solution compared to water intake in isolation, which shows a modest beneficial effect of beverages containing moderate to high amounts of sodium on pre-exercise fluid retention and hypervolemic response [194]. Of note, effects on water retention are unsurprising since drinks with high electrolyte content, such as ORS, have been shown to have higher BHI than commercial sports drinks and water [27,28,29].

Sodium-containing CES might increase fluid consumption through thirst, which may help lessen high pre-exercise hypohydration rates in athletes. In addition, these formulations may increase fluid retention and have mild hypervolemic effects; therefore, they may ameliorate dehydration during exercise, without the risk of possible adverse effects often observed with other hyperhydration protocols. Unfortunately, most research on the pre-exercise period has focused on maximizing glycogen stores to ensure adequate fuel supply during exercise, with only a few food products, such as coconut water, skimmed milk, non-alcoholic beer, and soup being explored to improve pre-exercise hydration status [195,196,197,198]. A rationale for the type, volume, and timing of pre-exercise hydration is not always clear, and further research on well-designed CES compositions to improve fluid balance before exercise is warranted.

### 4.2. During Exercise

In settings of moderate intensity (<70% maximal oxygen consumption rate, or VO_2_ max), short duration (<1 h) exercise performed under mild environmental conditions, there is typically no need for additional provisions of carbohydrates or water with electrolytes [199]. On the other hand, water may be insufficient to compensate for exercise challenges like fluid balance and fuel supply when bouts of exercise are repeated over a short-time period, and during high-intensity (>70% VO_2_max) or long-duration (>1 h) exercise, particularly in hot environments [199].

CES formulated to support physiological needs during higher intensity or longer duration types of exercise typically attempt to achieve three main objectives: restore fluid balance, minimize electrolyte losses, and support carbohydrate provisions. For the first one, controlling factors linked to gastric emptying and intestinal water absorption are key when formulating CES aimed at accelerating hydration during exercise. Among these, carbohydrate content and beverage osmolality are particularly important. As cited before, carbohydrates, such as glucose or galactose, co-transport with sodium to facilitate intestinal water absorption. However, CES with carbohydrate content >6% may delay gastric emptying, thus slowing effective fluid delivery to systemic circulation [91,92,93,94]. Of note, data on gastric emptying do not provide a complete picture of the processes involved in fluid delivery to systemic circulation, and recent position statements have suggested that formulations containing multiple transportable carbohydrates might help attenuate this issue through improved intestinal absorption while supporting higher rates of carbohydrate utilization when exercising in the heat [11]. Notwithstanding, delaying gastric emptying might lead to gastrointestinal distress and ineffective fluid replacement, which should be reasonably avoided in situations where rapid rehydration should be prioritized (i.e., exercise in the heat, athletes with high sweat rates).

To reiterate, the effect of beverage carbohydrate content on gastric emptying is attributed to its impact on both energy density and osmolality of the formulated solution, and CES osmolality also modulates intestinal water absorption through the regulation of luminal content osmolality. A recent systematic review and meta-analysis pooled data from 28 studies evaluating the impact of hypertonic (>300 mOsmol/kg), isotonic (275–300 mOsmol/kg), or hypotonic (<275 mOsmol/kg) CES on changes in plasma volume during continuous exercise. This review concluded that hypotonic drinks were more likely to maintain central hydration defined as a change in plasma volume, compared to isotonic CES, hypertonic CES, and water [200]. Electrolyte content was also shown to positively associate with improvements in plasma volume across hypotonic, isotonic, and hypertonic CES, albeit formulations containing >50 mmol/L sodium were excluded from analyses as these were proposed to have poor palatability, which might compromise ad libitum drinking [200]. It is noteworthy that recent research evaluating the sensory perception properties of ORS when consumed during exercise in the heat has reported increased overall liking with longer exercise duration [40]. Sweeteners, flavor additives, and types of carbohydrates are among the many aspects that can be tuned to improve the overall liking of CES containing moderate or high quantities of sodium (i.e., ≥45 mmol/L).

Regarding electrolyte loss, the ACSM 2007 position statement recommended consuming solutions containing 20–30 mmol/L sodium chloride during exercise depending on exercise duration, intensity, and environmental conditions [10]. In this sense, most sports drinks intended to replace sweat sodium losses are in the range of 10–25 mmol/L [201]. However, a 90 min soccer training session in cool temperature with a mean fluid intake of 423 mL has been shown to correlate to a sweat rate of 1.13 L/h and a total of 4.3 g NaCL loss (mean sweat sodium concentration of 42 mmol/L) in elite male soccer players [202]. Similar high sweat rates and sweat sodium concentrations have been reported in high-level soccer training and matches both under hot and cold environmental conditions [203,204,205]. Concerning different disciplines, a retrospective analysis of 1303 athletes observed that, despite important inter-individual differences, American football players and endurance athletes might experience even higher whole-body sweat rates and rates of sweat sodium loss compared to soccer players [206]. Although it is recommended to individualize sodium intake during exercise based on specific losses [2], it seems unclear whether CES containing 20–30 mmol/L of sodium is sufficient to match exercise-induced sodium loss in some scenarios [207]. Thus, CES containing sodium concentrations closer to the mid-range for sweat (i.e., 50 mmol/L) may serve the replacement needs of a greater proportion of athletes. Partially replacing electrolytes (particularly sodium) is important to maintain plasma osmolality, thus stimulating physiological thirst, and potentially attenuating or preventing EAMC during exercise [10].

When provided in solid form, salt supplementation has been shown to mitigate BML and may increase serum sodium and chloride levels during ultra-endurance events [208,209], which is important for preventing the onset of hyponatremia. Similarly, some authors have explored the role of sodium in liquid formulations for restoring fluid and electrolyte balance during long-lasting endurance exercise. In cyclists performing three 4 h rides at 55% VO_2_ peak in mild temperature, the consumption of a 100 mmol/L sodium beverage was associated with decreased urine production, and both 50 and 100 mmol/L sodium solutions had a mild impact on preventing decrements in plasma osmolality at 240 min compared to CES containing 5 mmol/L sodium [210]. In a recent study conducted by Wijering et al., 11 previously euhydrated males completed a 3 h cycling session at 55% VO_2_ max and 34 °C while consuming 6% carbohydrate CES with either 21 mmol/L or 60 mmol/L sodium every 15 min at a rate equal to fluid loss [211]. Authors observed that plasma sodium concentrations were significantly higher with the 60 mmol/L sodium beverage over the course of exercise compared with the 21 mmol/L sodium CES (change in plasma sodium from baseline to the end of the trial: 0.8 ± 2.4 mmol/L vs. −1.5 ± 2.2 mmol/L in the 60 mmol/L vs. the 21 mmol/L treatments, respectively) [211]. Further, plasma volume decreased over the course of the trial only in individuals consuming the 21 mmol/L sodium CES [211]. Of note, previous research using similar studies showed no difference between 21 mmol/L and 36 mmol/L sodium CES in maintaining serum sodium and plasma osmolality during exercise in the heat, which suggests that improvements in plasma sodium levels might be more evident with higher sodium intakes [212]. Finally, in a recent study conducted by Fan et al., the rehydrating effects of a hypertonic sports drink containing 6.2% carbohydrate, 31 mmol/L sodium, and 5.3 mmol/L potassium were compared to a hypotonic beverage containing 3.3% carbohydrate, 60 mmol/L sodium and 18.2 mmol/L potassium in healthy males [37]. Participants drank the CES during and after 75 min of cycling (65% VO_2_peak) in the heat at a volume equivalent to 150% BML, and authors reported that both CES were associated with similar ratings of palatability, and led to small increments in endurance performance during a second bout of exercise compared to water [37]. Nevertheless, cumulative urine volumes were significantly lower in the participants allocated to the hypotonic solution compared to the sports drink and water [37]. Overall, evidence from these studies suggests that CES with moderate to high sodium might help maintain plasma volume and sodium levels during exercise in the heat.

Due to its spontaneous and unpredictable onset, research on the etiology of EAMC has been shown largely unsuccessful. Early observations of coal miners showed that muscle cramping is more frequent when sweat rates are high, which suggests that fluid and electrolyte losses might induce EAMC [213]. Research on ultra-endurance athletes has not endorsed this hypothesis, suggesting that different mechanisms are involved [214]. However, reviews on the topic have proposed that different types of EAMC might respond to different mechanisms, including water and electrolyte imbalances and disturbed spinal reflex activity [215]. Regarding evidence on the first mechanism, Ohno et al. conducted a trial of nine young men with a previous history of EAMC. They were asked to isometrically flex their knee joint at a maximally shortened position for 15 s after dehydrating at 1%, 2%, or 3% BML through heat exposure [216]. Authors observed that three and six subjects developed EAMC in the 2% and 3% groups, respectively, while no cases were documented in the 1% group, which suggested an increased likelihood of EAMC with higher body fluid loss [216]. In a different study, supplementation of a sports drink with salt was observed to delay the onset of EAMC without having any positive impact on its prevalence in healthy males performing a calf-fatiguing protocol [217]. In a study conducted by Lau et al., men were dehydrated to 2% BML through downhill running in the heat [68]. Authors compared the effect of spring water and a hypotonic beverage containing 50 mmol/L sodium, 20 mmol/L potassium, and 1.8% glucose on the threshold frequency of electrical train stimulation (technique used to electrically induce muscle cramping through placing electrodes on muscles in the lower leg [218]) at 0, 30, and 60 min after ingestion observing that, while the threshold frequency decreased after water consumption, an increase was shown after CES intake at 30 and 60 min, respectively, which implies less susceptibility to the electrical simulation-induced muscle cramp [68]. Additionally, CES consumption helped maintain serum sodium levels while water intake did not [68]. In an ensuing study conducted by the same group, the same beverages were consumed during (instead of after) exercise in the heat, and similar results were observed at 65 min post-exercise [41]. Finally, another study conducted in euhydrated participants compared a CES containing 73 mmol/L sodium, 16.4 mmol/L potassium, 5.8% carbohydrate, and 0.6 g/L L-alanine to a placebo beverage containing a small amount of sodium (3 mmol/L), and reported an increase in the threshold frequency required to induce muscle cramping among individuals consuming the CES compared to those consuming the placebo beverage [69]. According to the authors, these results support electrolyte consumption influencing cramp susceptibility independent of hydration status [69]. Although there are some limitations inherent to the methods explored by these authors, such as the use of electrical stimulation to simulate muscle cramps, which does not fully replicate the natural occurrence of EAMC and its underlying mechanisms, CES used to restore electrolyte losses during exercise might help prevent or ameliorate EAMC, and this topic merits further research.

It is well-established that low muscle glycogen availability compromises the ability of muscles to exercise [219]. Since whole-body glycogen stores are limited (mostly stored in muscle (i.e., 500 g) and liver (i.e., 80 g)), carbohydrate provision during exercise is key to sustaining energy demands during high-intensity and/or longer-duration physical exercise [120]. Most expert institutions recommend 30–60 g/h of carbohydrates for endurance and intermittent sports of 1–2.5 h of duration, and up to 90 g/h for ultra-endurance events [10,12,120]. In this sense, CES are frequently used as a liquid vehicle for carbohydrate delivery during exercise. However, it should be noted that energy availability might not be the only limiting factor for endurance capacity in situations where fluid and electrolyte balance can be severely compromised, such as exercising in the heat or for athletes with high sweat rates [12]. Achieving 30–60 g/h of carbohydrates solely through CES intake at a volume of 0.4–0.8 L/h typically implies the use of standard isotonic or hypertonic drinks with carbohydrate content and associated osmolality that may slow gastric emptying, promote luminal water influx, and, in turn, compromise effective fluid delivery to the systemic circulation. Importantly, a few studies have shown that relatively small quantities of carbohydrates (i.e., 10–30 g/h) can still support endurance capacity [220,221,222]. Based on the abovementioned studies, CES formulations with low to moderate levels of carbohydrate (i.e., 2–6%) in combination with sufficient electrolyte content to match sweat losses might be more effective in sustaining exercise capacity in the heat. On the other hand, athletes might benefit from using formulations with higher quantities of carbohydrates during long-duration exercise (i.e., >2.5 h) in mild environmental conditions to support fuel needs. It is important to mention that some higher carbohydrate sports beverages (i.e., >6%) would not be primarily designed to optimize hydration or rehydration but to effectively support exercise fuel demands in situations where exercise dehydration is not the limiting factor for endurance capacity. Different scenarios in terms of exercise duration and intensity along with environmental conditions might require an individualized approach to hydration strategy based on trial-and-error experience.

### 4.3. After Exercise

In most situations, normal water and food intake practices can sufficiently restore fluid and electrolyte balance after exercise [223]. When a second bout of exercise is to be conducted within a short time period, athletes pursue recovering as fast as possible, with rehydration being an important part of the recovery process. To this end, the addition of electrolytes and carbohydrates to rehydration beverages for improving net fluid balance has been extensively explored.

In subjects dehydrated to 2.3% BML after exercising in the heat, the ad libitum intake of a 0.45% NaCl solution was shown to restore pre-dehydration plasma volume values within 20 min, while 60 min was needed when consuming water alone [224,225]. When a fixed quantity of fluid is provided (100% BML), a CES containing 6% carbohydrate and 20 mmol/L sodium was shown to restore both plasma volume and osmolality better than water or a low-electrolyte diet cola over a 2 h rehydration period in subjects previously dehydrated at 2.5% BML [226].

Subsequent studies have evaluated the differential effects of carbohydrate (glucose) and electrolyte types and amounts on post-exercise rehydration. Three formulations containing either 90 mmol/L glucose, 60 mmol/L sodium chloride, or 25 mmol/L potassium chloride were compared to a beverage containing all three ingredients (90 mmol/L glucose, 60 mmol/L sodium chloride, and 25 mmol/L potassium chloride). When the beverages were provided over 30 min to hypohydrated men (2% BML), it was shown that the addition of electrolytes to the rehydration beverages reduced cumulative urine volume compared to the solution containing only glucose after exercise, thus demonstrating improved net fluid balance [65]. Of note, both sodium and potassium displayed important fluid retention properties, yet the effects were not additive. In another study, the effects of drinks consumed at a volume of 150% BML and containing 2, 26, 52, or 100 mmol/L sodium for post-exercise rehydration in hypohydrated men (1.9% BML) were evaluated, and it was observed that urine volume over 5.5 h was inversely proportional to the amount of sodium provided [201]. Further, plasma volume was significantly higher only in subjects consuming the 52 and the 100 mmol/L sodium solutions at 1.5 h post-ingestion compared to the effects of the 2 mmol/L beverage [201]. Similarly, Shirreffs et al. compared the effects of 23 and 61 mmol/L sodium drinks consumed at 50%, 100%, 150%, and 200% BML on post-exercise rehydration in 2% dehydrated healthy adults, and concluded that, although a volume greater than sweat loss must be ingested to restore fluid balance, sufficiently high levels of sodium must also be provided to prevent increased urinary output alone [227]. Importantly, palatability was not an issue when consuming either of the tested beverages [227]. These results are aligned with those obtained by Merson et al., exploring four drinks containing 1, 31, 40, or 50 mmol/L sodium consumed at 150% BML in dehydrated men (≈2% BML). The authors observed greater cumulative volume of urine produced and poorer net fluid balance after 3–4 h with the 1 mmol/L compared to only the 40 and 50 mmol/L beverages [228]. In light of these studies, it seems clear that sodium at levels higher than those contained in typical isotonic or hypertonic sports drinks is beneficial for net fluid balance when rapid and effective rehydration is pursued. However, this is not clearly inferred from current expert guidelines on post-exercise rehydration, which are mainly focused on the role of solid food intake in restoring fluid and electrolyte balance [2,11,12], or simply state that fluids with sufficient electrolytes should be consumed over time [10].

Although previous studies showed that solutions containing glucose alone can lead to higher urine volume compared to electrolyte-containing solutions [65], other studies have reported a mild impact of carbohydrates on water retention when provided with standardized electrolyte provisions [71]. In this sense, men exercising in the heat to induce 2–3% BML were provided either a flavored placebo, a placebo containing electrolytes (18 mmol/L sodium and 3 mmol/L potassium), or one of three solutions containing electrolytes and 3%, 6%, or 12% sucrose and high fructose corn syrup combined [71]. While no difference was observed between the electrolyte-only beverage, and the 3% and 6% carbohydrate solutions, fluid retention was significantly higher with the 12% carbohydrate drink [71]. Similarly, after exercise-induced hypohydration, a 10% glucose solution was shown to be more effective in maintaining euhydration (1 h longer) than a 0% glucose beverage, when amounts of electrolytes were fixed [22]. Lastly, Kamijo et al. provided one of three solutions containing a fixed composition of electrolytes (21 mmol/L sodium, 5 mmol/L potassium, 16.5 mmol/L chloride, and 10 mmol/L citrate) and either 3.4% glucose + 3.1% fructose, 1.7% glucose + 1.6% fructose or no carbohydrate to men hypohydrated at 2.3% BML [229]. Aligned with previous studies, sodium retention was higher and plasma volume recovery was improved with increasing carbohydrate concentrations [229].

Evaluating different types of carbohydrates might help elucidate mechanisms underlying their fluid retention properties. For example, some slow-digesting carbohydrates, such as isomaltulose, have a lower glycemic index compared to sucrose (32 and 65, respectively [111]), and elicit a lower but more sustained insulinemic response [111]. Amano et al. tested the effects of three beverages containing only water, electrolytes (20 mmol/ sodium and 4.5 mmol/L potassium), and 20 g/L (2%), or 65 g/L (6.5%) of isomaltulose at a volume equal to BML for post-exercise rehydration in men dehydrated to 2% of previous body weight [230]. Authors observed significantly increased plasma volume recovery at 120 min along with higher plasma osmolality at 60 and 90 min in the 6.5% compared to the 2% isomaltulose trial [230]. Of note, there is also a report on euhydrated individuals that showed a higher BHI (indicated by attenuated urine production) for isomaltulose compared to sucrose and water, with the isomaltulose beverage also eliciting a delayed insulinemic response compared to the sucrose solution [231]. The current evidence is unclear whether the effects of slow-release carbohydrates, such as isomaltulose, on decreased urine output, can be attributed to a blunted and sustained insulinemic response or may be the consequence of a delay in gastric emptying or intestinal water absorption. Further research comparing different types and quantities of fast- and slow-digesting carbohydrates might shed light upon the mechanisms underlying the mild water retention effects of carbohydrates during rehydration after exercise.

As cited in previous sections, milk has been considered a more effective rehydration beverage than some common CES due to its improved effects on fluid retention [126,128,129,130]. Along with its electrolyte content, the effects of milk protein in delaying gastric emptying might be partially responsible for the effects of milk on net fluid balance [27]. Contrary to whey protein, milk protein isolate has been shown to decrease urine output compared to isoenergetic amounts of carbohydrate solutions with similar fat and electrolyte content [21,122]. Furthermore, adding whey protein to water does not decrease total urine volume, fluid retention, or net fluid balance during post-exercise rehydration [124], which highlights the role of the casein fraction in slowing gastric emptying. Delaying gastric emptying might be useful to help sustain high plasma osmolality over time and to help decrease urine output in situations where micturition is being avoided. However, in consideration of the science of hypertonic solutions, it is unclear whether slower gastric emptying is beneficial for achieving effective and rapid rehydration since it slows the expansion of plasma volume which, in turn, is necessary for maintaining adequate cardiovascular function and heat dissipation capacity in preparation for and during a second exercise bout.

In Table 3, a summary of the ways that CES formulations can optimize fluid and electrolyte balance before, during, and after exercise, as well as potential research gaps are presented. Additional gaps might include exploring potential novel ingredients, such as amino acids, in hydration beverages for exercise.

## 5. Outlook to Hypotonic Beverages in Sports

Science on CES compositions for fluid replacement has largely focused on ensuring sufficient carbohydrate provisions to sustain exercise performance and improving palatability to drive consumption. From a hydration point of view, sports drinks are also purported to accelerate fluid absorption and improve rehydration after exercise [33]. This is largely justified based on the intestinal cotransport of glucose and sodium to facilitate water uptake and the role of carbohydrates in retaining fluid after exercise. However, the levels of sodium contained in these CES are typically insufficient to match sweat sodium losses in high-intensity or long-duration scenarios, and the type or amounts of carbohydrates used to sustain fuel needs might not be optimal for effectively restoring plasma volume. Consequently, the use of ORS, with lower carbohydrate (i.e., 2–6%) and higher sodium content (i.e., ≥45 mmol/L), has gained momentum in recent years for improving hydration in exercise settings where compensating sweat losses should be prioritized [37,38,39,40,41]. Likewise, these hypotonic formulas are consistently reported to have higher BHI than sports drinks due to their elevated sodium content, which makes them an attractive choice in professional or clinical situations where fluid availability is limited or urine excretion should be avoided [27,28,30].

While ORS were originally intended to treat mild to moderate dehydration during acute diarrheal disease, these formulations might also be more effective than sports drinks for restoring fluid and electrolyte balance in some scenarios. First, sodium content ≥40 mmol/L may support exercise-induced sodium sweat losses and sustain plasma osmolality more effectively than lower sodium formulations (10–25 mmol/L) [37,201,211]. Further, solutions with lower osmolality (<275 mOsm/kg) have been consistently reported to optimize intestinal water uptake compared to isotonic and hypertonic formulations, along with plain water [200]. In addition, the presence of low amounts of carbohydrates (i.e., 2%) has been shown to accelerate intestinal water uptake without appearing to negatively impact gastric emptying or luminal osmolality. And, finally, these formulas contain moderate to high quantities of sodium, which is confirmed to play a key role in fluid retention during post-exercise rehydration [227]. All these aspects may make this type of CES a promising choice when evaluating compositions aimed at optimizing fluid and electrolyte balance in settings of high sweat rates or in situations where fuel supply is not the primary limiting factor for endurance capacity. In Table 4, available studies comparing the effects of ORS and sports drinks on hydration-related outcomes in exercise settings are summarized.

As shown in Table 4, not all studies have demonstrated these promising effects of ORS in exercise hydration. For example, a study comparing the hydration properties of an ORS (235 mOsm/L, 60.9 mmol/L sodium, 3.4% carbohydrates) and a sports drink (355 mOsm/L, 18.4 mmol/L sodium, 5.9% carbohydrate) consumed by ten adults walking in the heat (39 °C, 50% VO_2_ Max, 90 min) at a volume of 150% BML during the first 45 min failed to observe differences in percentage of dehydration or changes in plasma volume between treatments [38]. These types of discrepancies highlight the need for further studies evaluating similar CES compositions in different settings (i.e., high-intensity exercise, different training levels) [38]. Also, the addition of amino acids to ORS or sports drinks might offer additional benefits, as these are absorbed through sodium-dependent mechanisms as well as passive transport and sodium-independent transporters, which drive sodium and water absorption [28]. Combinations of carbohydrates and amino acids might produce potential additive effects, which is an interesting topic to explore in future studies.

## 6. Conclusions

Although extensively researched for decades, it remains unclear what compositions of carbohydrate–electrolyte solutions for fluid replacement in exercise can be considered optimal in different settings. While sports drinks can help maintain fuel supply and support the replenishment of glycogen stores during and after exercise, a growing body of literature suggests that these formulations might not be ideal for maintaining or restoring fluid and electrolyte balance in settings where large amounts of fluid and electrolyte losses are expected. Further, the levels of carbohydrates contained in these formulations might delay gastric emptying, and although this may not represent a complete picture of the appearance of beverages in the bloodstream, it should be reasonably avoided when pursuing fast and effective rehydration. As a consequence, recent research has attempted to explore hypotonic formulations with low carbohydrate (i.e., 2–6%) and moderate to high electrolyte content, mainly sodium (i.e., ≥45 mmol/L), to accelerate fluid delivery to systemic circulation, restore sweat fluid and electrolyte losses, and retain net fluid balance after exercise. Nonetheless, some comparisons between ORS and sports drinks have yielded mixed results, which highlight the need for further well-designed studies exploring different types of exercise intensity and duration, environmental conditions, sodium levels, and combinations of carbohydrate blends and additional ingredients to shedding light upon the role that these carbohydrate–electrolyte solutions can play in improving hydration before, during, and after exercise.

## Figures and Tables

**Figure 1 nutrients-16-00017-f001:**
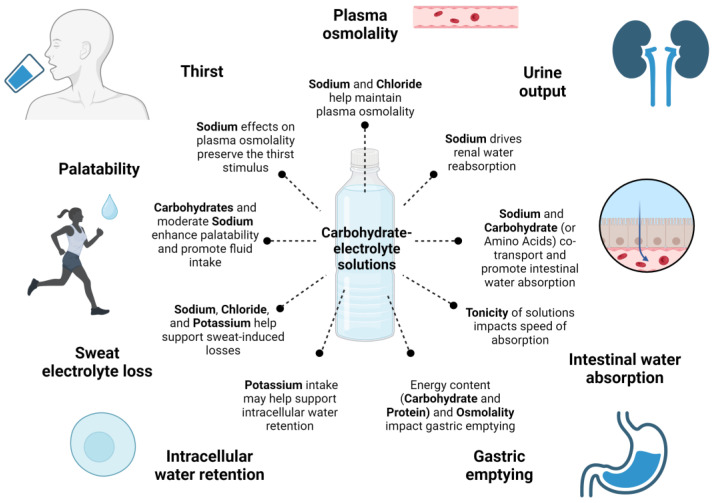
Impact of carbohydrate–electrolyte solution (CES) compositional aspects on hydration-related outcomes.

**Table 1 nutrients-16-00017-t001:** Recommendations from consensus documents on fluid intake plans (volume) for athletes.

Institution/Organization	Before Exercise	During Exercise	After Exercise
General			
American College of Sports Medicine (ACSM) (2007) [10]	Consume 5–10 mL fluid/kg body weight 2–4 h before exercise to allow sufficient time for voiding and achieve pale yellow color urine.	Consume sufficient fluid to replace sweat loss and limit BML to <2%. A plan suitable for most athletes may consist of consuming 0.4–0.8 L/h fluid during exercise.	Consume ≈ 1.5 L/kg BML at a modest rate to minimize urine loss.
National Athletic Trainers’ Association (NATA) (2017) [2]	Individualize fluid intake plans to achieve euhydration or <2% hyperhydration (the latter only under medical oversight, and before endurance events where fluid supply is limited).	Consume enough fluid to approximate personal sweat loss and limit BML to <2%.	Consume fluid up to 150% of estimated fluid loss in <4 h.
Hot environments			
Sports Dietitians Australia (SDA) (2020) [11]	Fluid intake strategies should be adapted to fluid balance and anticipated substrate and fluid requirements. Pre-event hyperhydration may be useful in situations of limited fluid intake opportunities.	Individualized fluid intake plans should be adjusted to real-time assessment and be based on prior fluid balance, thirst, gastrointestinal tolerance, and previous experience.	To rapidly reverse moderate–severe fluid deficit, volumes up to 150% of BML should be consumed during the hours following exercise.
International Olympic Committee (IOC) (2022) [12]	Fluid intake plans should ascertain sufficient fluid intake before training and competition in the heat to achieve BML < 1–2%, urine specific gravity < 1.020, and/or plasma osmolality < 290 mmol/kg.	Fluid intake plans should aim to minimize loss without increasing body weight during the event. Plans should be practiced at training under conditions similar to those of competition.	Rehydrate after exercise in the heat consuming fluids to restore fluid balance slightly over BML (i.e., 100–120%).

BML, body mass loss.

**Table 2 nutrients-16-00017-t002:** Compositional differences between standardized and commonly marketed oral rehydration solutions and sports drinks.

Carbohydrate–Electrolyte Solutions	Osmolality (mOsm/Kg)	Carbohydrate (Glucose) (g/L)	Sodium (mmol/L)	Potassium (mmol/L)	Chloride (mmol/L)
ORS (WHO, 1975) [45]	311	20	90	20	80
ORS (ESPGHAN, 1992) [46] *	200–250	13.3–20	50–60	20	60
ORS (WHO, 2002) [42]	245	13.5	75	20	65
Commonly Marketed ORS [36]	220–270	≤25	45–50	20–25	35–45
Commonly Marketed Sports Drinks [26]	>280–380	60–80	10–35	3–5	10–12

* Sodium levels updated in 2014 [47]. ESPGHAN, European Society for Paediatric Gastroenterology Hepatology and Nutrition; ORS, oral rehydration solution; WHO, World Health Organization.

**Table 3 nutrients-16-00017-t003:** Potential roles and research gaps of CES designed to optimize fluid and electrolyte balance before, during, and after exercise.

	Before Exercise	During Exercise	After Exercise
Rationale	Athletes might pursue commencing training and competition in a hyperhydrated state in situations of low fluid availability or when euhydration is insufficient to compensate for challenges in fluid balance [2].	Athletes typically want to prevent dehydration losses ≥2%, restore sweat electrolyte losses, or prevent EAMC during exercise [2].	Athletes might want to accelerate the restoration of fluid and electrolyte losses and reestablish net fluid balance before a further bout of exercise [2].
Evidence	Sodium loading has been shown to be more effective in increasing pre-exercise plasma volume than hyperhydration with water or glycerol [191].	Hypotonic formulations have been shown to absorb faster than isotonic or hypertonic formulas during exercise [200]. The amounts of sodium contained in isotonic or hypertonic sports drinks (10–25 mmol/L) may be insufficient to match sweat-induced losses in most situations [201]. Fluid and/or electrolyte imbalances might be associated with EAMC [215].	Sodium is the most important compositional aspect controlling urine production and retention of plasma volume when rehydrating after exercise [201]. Other macronutrients such as carbohydrates and proteins might promote water retention by delaying gastric emptying [21,82]. Carbohydrates might involve additional mechanisms (i.e., insulin response) [230].
Gaps in CES Research	Sodium levels at which CES produce a mild hypervolemic response and aid in fluid retention (i.e., ≥45 mmol/L) should be further researched. Palatability aspects of these formulations should also be considered. Most research to date has focused on maximizing glycogen stores, and further research on well-designed CES compositions to improve fluid balance before exercise is warranted.	Studies comparing isotonic/hypertonic sports drinks and hypotonic CES containing moderate to high amounts of sodium (i.e., ≥45 mmol/L) and low quantities of carbohydrate (i.e., 2–6%) and are scarce, and might shed light upon the role of these hypotonic solutions in aiding intestinal water absorption, restoring sweat electrolyte losses, and potentially preventing or attenuating EAMC during exercise. Raising sodium content of common isotonic/hypertonic sports drinks might help restore electrolyte balance during exercise, which merits further research. These studies might also benefit from exploring high-intensity exercise protocols to provide real-world evidence.	Further research on CES containing moderate to high amounts of sodium (i.e., ≥45 mmol/L) to help reduce urine output and increase plasma volume during post-exercise rehydration without compromising gastric motility is needed. Studies exploring slow- and fast-digestible carbohydrates might help elucidate mechanisms involved in water retention properties of carbohydrate-containing beverages.

BHI, beverage hydration index; CES, carbohydrate–electrolyte solutions; EAMC, exercise-associated muscle cramps.

**Table 4 nutrients-16-00017-t004:** Studies comparing the effects of oral rehydration solutions and sports drinks on hydration in exercise settings.

Reference	Study Design and Sample	Beverages	Exercise Protocol	Fluid Intake	Main Results
Schleh et al., 2018 [38]	Randomized double-blind cross-over trial *n* = 10 aerobically fit men	ORS: 235 mOsm/L; 60.9 mmol/L Na^+^; 3.4% carbohydrate; 20 mmol/L K^+^. Sports drink: 355 mOsm/L; 18.4 mmol/L Na^+^; 5.9% carbohydrate; 3.2 mmol/L K^+^.	Walking (50% VO_2_max, 90 min, 39 °C)	150% BML volume (calculated in the first 45 min) provided during exercise	Similar changes in percentage of dehydration, urine specific gravity, urine volume, and plasma volume were observed between treatments.
Fan et al., 2020 [37]	Randomized double-blind cross-over trial *n* = 9 physically active men	ORS: 216 mOsm/L; 60 mmol/L Na^+^; 3.3% carbohydrate; 18.2 mmol/L K^+^. Sports drink: 382 mOsm/L: 31 mmol/L Na^+^; 6.2% carbohydrate; 5.3 mmol/L K^+^. Water.	Cycling (65% VO_2_max, 75 min, 30.4 °C)	150% BML volume provided during exercise and 2 h post-exercise	Cumulative urine output over 5 h of recovery was lower and percentage fluid retention was higher with ORS than with sports drink and water. Serum sodium levels were maintained better with ORS than with sports drinks and water at 3 h of recovery. Net fluid balance, changes in plasma volume, skin temperature, and palatability ratings were similar across treatments.
Ly et al., 2023 [232]	Randomized double-blind cross-over trial *n* = 26 physically fit men	ORS: 270 mOsm/Kg; 45 mmol/L Na^+^; 2.5% carbohydrate; 20 mmol/L K^+^. Sports drink: 330–380 mOsm/Kg; 18 mmol/L Na^+^; 6% carbohydrate; 3 mmol/L K^+^. Water placebo.	Interval training performed until 2.6% BML (intermittent and variable intensity, 90 min, 22–29 °C)	100% BML volume provided 45 min post-exercise	Fluid retention at 3.5 h post-exercise was similar between ORS and sports drink, but urine volume was significantly reduced with the ORS at 30–60 min into recovery, which was associated with improved sodium balance.

BML, body mass loss; K^+^, potassium; Na^+^, sodium; ORS, oral rehydration solution; VO_2_max, maximal oxygen consumption rate.

## Data Availability

Data sharing not applicable.

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
