# Peer review of "Compositional Aspects of Beverages Designed to Promote Hydration Before, During, and After Exercise: Concepts Revisited"

_nutrients, 2023, doi:10.3390/nu16010017_

Round 1
Reviewer 1 Report
Comments and Suggestions for Authors
This review paper provides a fairly comprehensive (although not technically a systematic review) reference list of papers related to hydration research before, during and after exercise, including several position stands that various scientific organizations have published on the topic. It is a rather comprehensive document for which the authors should be commended. However, due to the breadth of the topic there remains several important statements that lack references throughout. Therefore, the question remains: what is the goal of this review and what is new? The purpose is vaguely stated. The paper might be improved by focusing specifically on one context (either before, during or after exercise) to develop more specificity in identifying potential gaps in the literature (or contradictory findings). It seems like this review paper is specifically questioning whether ORS formula are “superior” to the recommendations of sports drinks in the literature but dances around this point. Therefore, a clear delineation what are these two classes of beverages needs to be made in a referenced Table (along with changes in WHO ORS changes over the years). The last section, role of hypotonic beverages in sport is the most thought-provoking part of the paper. The section of potential other ingredients might be expanded on amino acids as the emerging area. Caffeine during exercise is performance enhancing and there have been many more papers for its use during exercise in the heat than cited. It may be advantageous to limit the discussion of caffeine and alcohol. Most of the focus on the caffeine section is consumption throughout a typical day and not necessarily exercise-relevant. Suggest narrowing the focus to revisiting optimal ingredients for ORS vs. Sport Beverages (and much more elaboration with Data Summary or Evidence tables for study findings) would improve the significance of the manuscript.
The main limitation of what is the stated goal of this review paper given there are numerous Position stands that already exist. The abstract and manuscript indicates “no consensus on optimal formulations” for hydration beverages which seems to be a flawed premised based on the countless evidenced-based reviews on the subject. The conclusion that ‘a consensus on the optimal compositions of carbohydrate-electrolyte solutions for fluid replacement in exercise has not been reached” is not well supported based on the literature base which has been extensively studied and reviewed. The conclusions go on to rehash older concepts related to gastric emptying that do not fully present the full picture for appearance of beverages into the bloodstream. This paper also fails to define what is “low” versus “moderate to high electrolytes”.
Specific points:
Title might consider either “Future directions or Concepts revisited”?
The abstract (line 20-21) is inaccurate and unsupported as several Position stands are cited in the paper. Similarly for line 23-24- the old dogma that carbohydrate is not ideal for long duration in the heat is largely reliant on older papers limited to gastric emptying (none using D2O).
Line 29 The conclusion is weakly stated “future studies might benefit”. Instead, focus on the gaps in the literature or areas in these broad contexts of fluid replacement where there might be room to further our understanding, particularly if the drink formula clearly needs to change in post-ex compared to during exercise.
Line 121 That carbohydrate is “out of the scope of this review” is clearly odd. Then the purpose to not comprehensively collect evidence for optimal CES but provide rational of available ingredients seems like a contradictory statement. As such, the goal of this paper becomes murky.
Line 133 Provide more information regarding the search string used. Are non-published theses from ProQuest included?
Line 165- defend the statement more specifically made “guidelines with loose recommendations”.
The section 3 should have a comparison table of ORS vs. CE sport drinks composition and studies that have compared them. The question becomes are these interchangeable for various functionality (pre, during or post).
Line 178- Ref 51 is specific to “rehydration” per se.
Line 186 – Most critical “aspect” for CES or is it the most critical electrolyte?
Line 204- Role of potassium and EAMC needs some referencing.
Line 220- some points of potential benefit of Mg+ and EAMC should be added (or at least what evidence is there) since magnesium sulfate is often a treatment and there is a Cochrane review on the subject.
Line 234 References required for this statement (fructose to glucose ratio >1 discouraged). Where is the evidence? There are some recommendation documents (Jeukendrup and Baker review) that have variable carbohydrate recommendations depending on duration etc. These appear to be missing here.
Section 3.2 on Carbohydrates fails to include classic paper by Jeukendrup 2009 showing the D2O appearance (thus reflecting both gastric emptying and intestinal absorption processes) for 0-~9% carbohydrates. The entire section only describes older gastric emptying studies, thus inadequately stating the rationale for limiting carbohydrate content. In fact, there is no mention whatsoever in the background information(first 8 pages) of relative fluid delivery with more recent D2O techniques (that reflect both GE and intestinal absorption) and what these papers indicate (but mention one review of Tripe-lumen papers published 13 years ago).
Table 1. The ACSM Position Stand on fluid replacement is not referenced (41) and should be replaced in Table 1 with the specific points which is a more comprehensive and focused than the Joint Nutrition Position Stand cited. I believe there is also a revision under way.
Table 2 requires references to support the major points made.
Line 437 – “extensively” utilized- seems a bit unsubstantiated based on 2 studies and is this (e.g. glycerol) recommended in the 153 consensus?
Line 527. The gastric emptying is just one aspect of fluid delivery that is not always valid in actual exercise studies in the heat. Hence, the ACSM fluid replacement position statement adjusts this older dogma accordingly.
Lines 548-565 The need to fully replace sodium losses during exercise does not seem to be rationally presented. There is ample opportunity to replace sodium post-exercise. There is no discussion of the level of sodium from that defined by the EU (and other global bodies) as a “sports drink” formula. The phrase…“any potential surplus should not pose any harm”- where is the evidence for this statement?
Line 596 Define what moderate-to-high sodium is.
The section on electrolytes during exercise (re-visiting the optimal concentration) was interesting but again a table would potentially be beneficial summarize evidence for different concentrations. Table 2 does not float the possibility of raising sodium in isotonic or hypertonic as paper by Jeukendrup showed sodium concentration does not impact fluid availability (using D20 technique). Perhaps the Table 2 should instead point out potential gaps in knowledge and not try to re-state recommendations as its goal.
Line 659 Define what “higher-carbohydrate” sports beverage is.
Line 682 Study design is unclear as stated.
Line 705. Provide the recommendations listed in those documents.
Author Response
Reviewer 1
Comments and Suggestions for Authors
- This review paper provides a fairly comprehensive (although not technically a systematic review) reference list of papers related to hydration research before, during and after exercise, including several position stands that various scientific organizations have published on the topic. It is a rather comprehensive document for which the authors should be commended.
We deeply appreciate the reviewer’s thorough examination of our manuscript, and we are thankful for the constructive feedback provided. We have carefully considered each comment with the aim of improving the quality of our manuscript, and provided responses to all the raised concerns.
- However, due to the breadth of the topic there remains several important statements that lack references throughout.
We thank the reviewer for pointing out this issue. In line with comments 22, 25, and 37, we have revised the manuscript and added citations to all important statements.
- Therefore, the question remains: what is the goal of this review and what is new? The purpose is vaguely stated. The paper might be improved by focusing specifically on one context (either before, during or after exercise) to develop more specificity in identifying potential gaps in the literature (or contradictory findings).
We agree that the research goal was not sufficiently clear. In line with comments 7 and 14, we adapted the goal in lines 128-131 (highlighted) to narrow the focus to revisiting optimal ingredients for CES to address hydration and dehydration before, during, and after exercise as well as to review the most recent advancements in the field. We consider that the aim is clearer now and consistent with the rest of the manuscript.
While we agree that focusing on one specific time-point (i.e., post-exercise) might be useful to develop a more specific review aimed to better understand potential gaps in the literature, it was our original intention to approach the topic from a broader point of view, since we believe that important advancements have been made across a wide range of CES compositions designed for improving pre-exercise, exercise, and post-exercise hydration/rehydration. Further, we believe that the hydration properties of hypotonic formulas have been overlooked in exercise literature across a range of situations, including pre-, during and post-exercise settings.
- It seems like this review paper is specifically questioning whether ORS formula are “superior” to the recommendations of sports drinks in the literature but dances around this point. Therefore, a clear delineation what are these two classes of beverages needs to be made in a referenced Table (along with changes in WHO ORS changes over the years).
We appreciate the suggestion. A new table (Table 2) was added to section 3 (lines 168-173) (highlighted) to clarify the compositional differences of standardized ORS and commercial ORS and sports drinks.
- The last section, role of hypotonic beverages in sport is the most thought-provoking part of the paper. The section of potential other ingredients might be expanded on amino acids as the emerging area.
We thank the reviewer for the recommendation. Accordingly, we have expanded section 3.4 adding new references, and highlighting amino acid-based formulas as an emerging field of research (lines 406-418) (highlighted).
- Caffeine during exercise is performance enhancing and there have been many more papers for its use during exercise in the heat than cited. It may be advantageous to limit the discussion of caffeine and alcohol. Most of the focus on the caffeine section is consumption throughout a typical day and not necessarily exercise-relevant. Suggest narrowing the focus to revisiting optimal ingredients for ORS vs. Sport Beverages (and much more elaboration with Data Summary or Evidence tables for study findings) would improve the significance of the manuscript.
As the reviewer mentions, there are more papers documenting the use of caffeine in hot environments as an ergogenic compound. In an attempt to summarize this information, we cited the systematic review and meta-analysis conducted by Zhang et al., in 2015, where many of these articles are synthetized. While some of the papers we cited in this section explored caffeine consumption throughout a typical day, these have permitted the study of the mechanisms involved in its diuretic effects at rest, which led to following research aimed to evaluate if these are maintained during exercise. Although we share the opinion that caffeine, and particularly alcohol, might not be considered as “optimal ingredients” for beverages designed to improve hydration and dehydration in exercise, we consider that there is sufficient interest in the literature to discuss their potential application to these formulas. Lastly, in line with reviewer’s recommendations, we provided a new table (Table 4) (lines 862-863) (highlighted) where data on studies comparing ORS and Sports drinks were summarized, including a recent study that was not previously included. Of note, while there is increasing interest in the role of ORS in sports, comparisons with typical sports drinks are still scarce.
- The main limitation of what is the stated goal of this review paper given there are numerous Position stands that already exist. The abstract and manuscript indicates “no consensus on optimal formulations” for hydration beverages which seems to be a flawed premised based on the countless evidenced-based reviews on the subject. The conclusion that ‘a consensus on the optimal compositions of carbohydrate-electrolyte solutions for fluid replacement in exercise has not been reached” is not well supported based on the literature base which has been extensively studied and reviewed.
We agree that numerous reviews and position stand documents on optimal formulations for hydration beverages have been published over the years, and that stating that no consensus has been reached on this topic might lead to confusion. Thus, we clarified these aspects wherever mentioned (Abstract, Conclusions, and Introduction (Lines 20-21; 101-103; 880-882) (highlighted). On this subject, we are of the opinion that previous reviews failed to provide precise recommendations on these formulations, oftentimes justified by the role that environmental conditions, athlete level and acclimatization, and exercise settings play in modulating exercise-related hydration demands. For example, the ACSM position stand on fluid replacement (cited in comment 24) states for pre-exercise hydration “consuming beverages with sodium (20-50 mEq·L−1) and/or small amounts of salted snacks or sodium-containing foods at meals will help to stimulate thirst and retain the consumed fluids”. We believe that concentrations closer to the upper limit might be more beneficial for retaining fluid / enhancing volemia before exercise, as presented in our work, and this is not clearly inferred from available literature. Despite the wealth of reviews and position stand documents, we consider that there remain uncertainties with regard to optimal compositions for improving hydration before, during, and after exercise, as suggested by the increasing interest in the role that ORS might play in this field.
- The conclusions go on to rehash older concepts related to gastric emptying that do not fully present the full picture for appearance of beverages into the bloodstream.
We specified that gastric emptying does not provide a full picture of the appearance of consumed beverages into the bloodstream in lines 897-888 (highlighted). Nonetheless, in line with comments 23 and 27, we believe that data on gastric emptying is still valuable to this date, as denoted by the heterogenous results yielded in studies using D2O methodologies (lines 277-295) (highlighted), and recommendations provided by consensus documents “If carbohydrates are included in the fluid, the optimal concentration for fluid absorption is between 3% and 8%, but concentrations. greater than 5% to 8% may slow the rate” [1] (references provided at the end of this document).
- This paper also fails to define what is “low” versus “moderate to high electrolytes”.
We thank the reviewer for pointing this out. We defined low and moderate to high amounts of carbohydrates and electrolytes the first time they appeared in each section throughout the manuscript and tables.
- Specific points:
Title might consider either “Future directions or Concepts revisited”?
We appreciate the suggestion, and changed the title accordingly (line 3) (highlighted).
- The abstract (line 20-21) is inaccurate and unsupported as several Position stands are cited in the paper.
In line with comment 7, we have clarified this sentence (lines 20-21) (highlighted).
- Similarly for line 23-24- the old dogma that carbohydrate is not ideal for long duration in the heat is largely reliant on older papers limited to gastric emptying (none using D2O).
We acknowledge that long-duration exercise might benefit from higher carbohydrate intake depending on the characteristics of the exercise performed, and adapted the sentence accordingly (line 24). Of note, we consider that (rather than carbohydrates) isotonic/hypertonic sports drinks might contribute to fluid balance, but these might not be “ideal” for effectively and rapidly restoring fluid and electrolyte balance during exercise in the heat, as shown by recent research exploring ORS formulas, which might represent interesting alternatives.
- Line 29 The conclusion is weakly stated “future studies might benefit”. Instead, focus on the gaps in the literature or areas in these broad contexts of fluid replacement where there might be room to further our understanding, particularly if the drink formula clearly needs to change in post-ex compared to during exercise.
We appreciate the suggestion, and changed the conclusion in the Abstract accordingly (lines 28-31) (highlighted). While the rationale for CES compositions might change based on the timing when these are consumed (as discussed in section 4), we believe that the main conclusions of our work should be focused on the role that hypotonic formulations might play in addressing hydration and rehydration before, during, and after exercise, which has not been specifically reviewed before.
- Line 121 That carbohydrate is “out of the scope of this review” is clearly odd. Then the purpose to not comprehensively collect evidence for optimal CES but provide rational of available ingredients seems like a contradictory statement. As such, the goal of this paper becomes murky.
In line with comments 3 and 7, we agree that the research goal was not sufficiently clear. Thus, we changed the goal and removed cited text line.
- Line 133 Provide more information regarding the search string used. Are non-published theses from ProQuest included?
While a comprehensive search was conducted, we did not use a systematic approach. Therefore, we did not provide a complete search strategy including search strings used in the databases, as we do not expect our search to be replicable and several records were retrieved from reference lists using a snowball strategy. Nonetheless, in line with reviewer’s suggestion, we changed the Materials and Methods section to provide more information about some of the search strings used. We cited two examples, including one that effectively identifies articles comparing ORS and sports drinks in ProQuest Dialog (sorted by relevance): ("Oral Rehydration Solution") OR ("Sports Drink") OR ("Sports Beverage") AND "Exercise". Used strings provided a large number of results; thus, an initial search was conducted in February 2023 and was updated periodically (sorting by date). We did not include non-published theses from ProQuest, as only Embase and PubMed (Medline) were selected (ProQuest Dissertations and Theses Professional database was excluded). We clarified this aspect in lines 137-139 and 140-142 (highlighted).
- Line 165- defend the statement more specifically made “guidelines with loose recommendations”.
We thank the reviewer for pointing this out. In line with comment 7, we cited the ACSM position stand on fluid replacement (line 177) (highlighted).
- The section 3 should have a comparison table of ORS vs. CE sport drinks composition and studies that have compared them. The question becomes are these interchangeable for various functionality (pre, during or post).
We appreciate the input received. In line with comment 4, we provided a table in section 3 (Table 2) (lines 168-173) (highlighted) where compositions of ORS and Sports drinks were presented. Another table (cited in comment 6) (lines 862-863) (highlighted) was presented in section 4 to summarize findings of studies comparing ORS and sports drinks.
- Line 178- Ref 51 is specific to “rehydration” per se.
We changed the citation to a general reference on the topic (line 190).
- Line 186 – Most critical “aspect” for CES or is it the most critical electrolyte?
We agree that the suggestion is more accurate and changed the sentence accordingly (line 198) (highlighted).
- Line 204- Role of potassium and EAMC needs some referencing.
We thank the reviewer for bringing this to our attention. We added two references of studies evaluating potassium-containing CES to restore electrolyte balance, which the authors suggested might help to prevent EAMC (line 217).
- Line 220- some points of potential benefit of Mg+ and EAMC should be added (or at least what evidence is there) since magnesium sulfate is often a treatment and there is a Cochrane review on the subject.
We read with interest a recent update (2020) of a Cochrane review on magnesium for skeletal muscle cramp [2]. A total of 11 studies evaluating magnesium-based interventions were identified, but none of them investigated exercise-associated cramps [2]. Further, we screened scientific literature on EAMC from 2020 to this date, and only found a recent article reporting a survey-based study where a magnesium-containing “electrolyte mix” was provided to half-marathon runners [3]. While the authors argued that the mix was effective for reducing mild to severe EAMC in this population, methodological concerns deterred us from citing this research [3]. We cited the mentioned Cochrane review to highlight the lack of well-conducted research (lines 233-236) (highlighted) (references provided at the end of this document).
- Line 234 References required for this statement (fructose to glucose ratio >1 discouraged). Where is the evidence? There are some recommendation documents (Jeukendrup and Baker review) that have variable carbohydrate recommendations depending on duration etc. These appear to be missing here.
We appreciate reviewer’s feedback on this matter. We provided a new reference to support the statement. In this study, fructose malabsorption was only present when fructose-glucose ratios exceeded 1. Also, we cited a review conducted by Rowlands et al., where studies comparing provisions of carbohydrates consisting of different fructose:glucose ratios are analyzed. In this review, almost all studies conducted to date explore fructose to glucose/maltodextrin rations ≤1 [4] (lines 251-253) (highlighted).
To the best of our knowledge, most research on optimal glucose-fructose ratios has focused on achieving higher carbohydrate oxidation rates (exceeding 60 g/h), which is reasonably impacted by intestinal absorption and liver/splanchnic metabolism. As discussed by Baker and Jeukendrup [5], a 0.5 ratio fructose:glucose (stated as 2 ratio glucose:fructose) is often recommended for athletes to achieve a 90g/h rate. Ingesting multiple transportable carbohydrates might be particularly beneficial, in terms of performance, for long duration exercise, where relatively high intakes are needed (> 60 g/h) [5]. However, optimal ratios for promoting carbohydrate oxidation rates might be out of the scope of our work, which is focused on the impact that these compositions have, specifically, on water balance in exercise settings (references provided at the end of this document).
- Section 3.2 on Carbohydrates fails to include classic paper by Jeukendrup 2009 showing the D2O appearance (thus reflecting both gastric emptying and intestinal absorption processes) for 0-~9% carbohydrates. The entire section only describes older gastric emptying studies, thus inadequately stating the rationale for limiting carbohydrate content. In fact, there is no mention whatsoever in the background information(first 8 pages) of relative fluid delivery with more recent D2O techniques (that reflect both GE and intestinal absorption) and what these papers indicate (but mention one review of Tripe-lumen papers published 13 years ago).
We deeply appreciate this input. In fact, we overlooked the classic paper published by Jeukendrup et al., in 2009 and changed section 3.2 accordingly to present their results and provide background on D2O techniques (lines 277-295) (highlighted). Of note, we believe that, while D2O techniques provide insightful data on unidirectional fluid flow, D2O enrichment does not represent a complete picture of the changes in body water that occur after the consumption of beverages with different carbohydrate or electrolyte content. This is exemplified by the heterogenous results reported in different studies in terms of enrichment in plasma D2O (i.e., ref [6-8]). Thus, we consider that this information might not always replace data on changes in plasma volume and gastric emptying (references provided at the end of this document).
- Table 1. The ACSM Position Stand on fluid replacement is not referenced (41) and should be replaced in Table 1 with the specific points which is a more comprehensive and focused than the Joint Nutrition Position Stand cited. I believe there is also a revision under way.
We agree that the cited Position Stand is more comprehensive, and its recommendations remain similarly unchanged to this date. Thus, we replaced the citation and adapted Table 1 accordingly (line 77) (highlighted).
- Table 2 requires references to support the major points made.
We added references to all major points made in the table.
- Line 437 – “extensively” utilized- seems a bit unsubstantiated based on 2 studies and is this (e.g. glycerol) recommended in the 153 consensus?
We agree that “extensively” was unsubstantiated based on the number of studies reviewed, and removed the word (line 487). In fact, only sodium-containing beverages were mentioned in the cited consensus statement. We appreciate the input.
- Line 527. The gastric emptying is just one aspect of fluid delivery that is not always valid in actual exercise studies in the heat. Hence, the ACSM fluid replacement position statement adjusts this older dogma accordingly.
We acknowledge that recent position statements on exercise in the heat have mentioned that “previous concerns” on the impact of exercise beverages on gastric emptying can be addressed through changes in the formulations, namely the use of multiple transportable carbohydrates, which might help address the higher rates of carbohydrate utilization when practical exercise in the heat [9]. We changed the mentioned line to adjust this information (lines 577-582) (highlighted). Nonetheless, we believe that, while aspects involved in fluid absorption throughout the gastrointestinal track should be taken into account as well, gastric emptying can be important for fluid replacement when exercising in the heat. As mentioned by Baker and Jeukendrup, “when both energy provision and fluid replacement are important (e.g., exercise in a warm environment), the optimal beverage would be formulated to deliver nutrients to the body without impeding gastric emptying and intestinal absorption of water” [5] (references provided at the end of this document).
- Lines 548-565 The need to fully replace sodium losses during exercise does not seem to be rationally presented. There is ample opportunity to replace sodium post-exercise.
We thank the reviewer for this comment. Controversy exists over the need of fully replacing electrolytes during exercise, as there are opportunities for replacing sodium post-exercise in many situations. However, that should not underestimate the importance of at least partially replacing electrolytes lost in sweat during exercise, which is key for maintaining a physiological drive to drink, and potentially attenuating or preventing exercise-associated muscle cramps. We highlighted these aspects in lines 620-622 (highlighted).
- There is no discussion of the level of sodium from that defined by the EU (and other global bodies) as a “sports drink” formula.
This is a particularly interesting comment. There is no formal legal definition of CES, as these are classified as food products by both US and European regulations. In Europe, the European Food Safety Authority (EFSA) recommends that these solutions should contain 20-50 mmol/L (0.46-1.15 g/L) sodium, provide 80-350 kcal/L, with at least 75% of the energy derived from carbohydrates that have high-glycemic response, and with osmolality between 200-330 mOsm/kg [10,11]. Naturally, both sports drinks and oral rehydration solutions (ORS) should be considered CES as carbohydrates and electrolytes are key ingredients in their formulations. However, compositional ranges recommended by the EFSA are aimed to provide industry guidance and do not necessary represent what can be considered optimal for fluid replacement (or other purposes). After much deliberation, we consider that citing these regulations might induce confusion to the reader (for example, standardized ORS might be considered CES or not depending on US or European regulatory frameworks), thus we opted for not including this information. Nonetheless, we are willing to re-adapt if considered needed (references provided at the end of this document).
- The phrase…“any potential surplus should not pose any harm”- where is the evidence for this statement?
While we are not aware of cases reported in the literature of athletes suffering adverse events associated with the ingestions of the electrolyte-containing beverages mentioned in this section due to electrolyte content, no studies have demonstrated otherwise, thus we removed the sentence (line 620).
- Line 596 Define what moderate-to-high sodium is.
We defined these levels throughout the manuscript, in line with comment 9.
- The section on electrolytes during exercise (re-visiting the optimal concentration) was interesting but again a table would potentially be beneficial summarize evidence for different concentrations.
We appreciate the suggestion. While we conducted a comprehensive search, we did not use a systematic approach, thus we cannot ensure that this type of table would contain all studies conducted to date exploring ranges of sodium during exercise. We decided keeping a narrative approach, but we are willing to synthetize the information in this section if the reviewer still considers there is value in it.
- Table 2 does not float the possibility of raising sodium in isotonic or hypertonic as paper by Jeukendrup showed sodium concentration does not impact fluid availability (using D20 technique).
We appreciate this very interesting suggestion. We added this information as a potential gap in knowledge, in line with comment 34 (lines 824-825) (highlighted). Of note, Jeukendrup failed to find differences in fluid delivery when comparing 6% glucose and 6% glucose + 20 mmol/L sodium, which should not be expected based on the literature on carbohydrate-electrolyte co-transport. The authors justified these findings by the presence of 6% carbohydrate, which may have masked fluid delivery [12] (references provided at the end of this document).
- Perhaps the Table 2 should instead point out potential gaps in knowledge and not try to re-state recommendations as its goal.
We agree that this approach would be more informative, and changed the table accordingly. We did not add citations to these new statements as these should be considered as research gaps, yet we are willing to re-adapt if needed (lines 824-825) (highlighted).
- Line 659 Define what “higher-carbohydrate” sports beverage is.
We added example ranges for all these definitions throughout the manuscript, in line with comment 9 (line 716).
- Line 682 Study design is unclear as stated.
We agree that the study design was not clear enough. We rewrote the sentence to improve clarity (lines 736-744) (highlighted).
- Line 705. Provide the recommendations listed in those documents.
We clarified this sentence citing the recommendation provided by the listed documents (lines 765-767) (highlighted). Besides the ACSM position stand, cited documents only reference the role of sodium-containing food intake after exercise, without further elaborating on the compositional aspects of CES designed for rapid rehydration during post-exercise recovery. While the importance of food intake in restoring electrolyte balance is undoubtable, CES can contribute to the recovery process in situations where the athlete aims to rehydrate as fast as possible.
References
- McDermott, B.P.; Anderson, S.A.; Armstrong, L.E.; Casa, D.J.; Cheuvront, S.N.; Cooper, L.; Kenney, W.L.; O'Connor, F.G.; Roberts, W.O. National Athletic Trainers' Association Position Statement: Fluid Replacement for the Physically Active. 2017.
- Garrison, S.R.; Korownyk, C.S.; Kolber, M.R.; Allan, G.M.; Musini, V.M.; Sekhon, R.K.; Dugré, N. Magnesium for skeletal muscle cramps. Cochrane Database of Systematic Reviews 2020, doi:10.1002/14651858.CD009402.pub3.
- Kharait, S. A Magnesium-Rich Electrolyte Hydration Mix Reduces Exercise Associated Muscle Cramps in Half-Marathon Runners: Direct Original Research. Journal of Exercise and Nutrition 2022, 5.
- Rowlands, D.S.; Houltham, S.; Musa-Veloso, K.; Brown, F.; Paulionis, L.; Bailey, D. Fructose-Glucose Composite Carbohydrates and Endurance Performance: Critical Review and Future Perspectives. Sports Med 2015, 45, 1561-1576, doi:10.1007/s40279-015-0381-0.
- Baker, L.B.; Jeukendrup, A.E. Optimal composition of fluid-replacement beverages. Compr Physiol 2014, 4, 575-620, doi:10.1002/cphy.c130014.
- Davis, J.M.; Burgess, W.A.; Slentz, C.A.; Bartoli, W.P. Fluid availability of sports drinks differing in carbohydrate type and concentration. Am J Clin Nutr 1990, 51, 1054-1057, doi:10.1093/ajcn/51.6.1054.
- Koulmann, N.; Melin, B.; Jimenez, C.; Charpenet, A.; Savourey, G.; Bittel, J. Effects of different carbohydrate-electrolyte beverages on the appearance of ingested deuterium in body fluids during moderate exercise by humans in the heat. 1997.
- Currell, K.; Urch, J.; Cerri, E.; Jentjens, R.L.; Blannin, A.K.; Jeukendrup, A.E. Plasma deuterium oxide accumulation following ingestion of different carbohydrate beverages. Applied physiology, nutrition, and metabolism = Physiologie appliquee, nutrition et metabolisme 2008, 33, 1067-1072, doi:10.1139/h08-084.
- McCubbin, A.J.; Allanson, B.A.; Caldwell Odgers, J.N.; Cort, M.M.; Costa, R.J.S.; Cox, G.R.; Crawshay, S.T.; Desbrow, B.; Freney, E.G.; Gaskell, S.K.; et al. Sports Dietitians Australia Position Statement: Nutrition for Exercise in Hot Environments. International Journal of Sport Nutrition and Exercise Metabolism 2020, 30, 83-98, doi:10.1123/ijsnem.2019-0300.
- EFSA Panel on Dietetic Products, N.; Allergies. Scientific Opinion on the substantiation of health claims related to carbohydrate‐electrolyte solutions and reduction in rated perceived exertion/effort during exercise (ID 460, 466, 467, 468), enhancement of water absorption during exercise (ID 314, 315, 316, 317, 319, 322, 325, 332, 408, 465, 473, 1168, 1574, 1593, 1618, 4302, 4309), and maintenance of endurance performance (ID 466, 469) pursuant to Article 13 (1) of Regulation (EC) No 1924/2006. EFSA Journal 2011, 9, 2211.
- SCF. Report of the Scientific Committee on Food on composition and specification of food intended to meet the expenditure of intense muscular effort, especially for sportsmen. European Commission, Brüssel 2001.
- Jeukendrup, A.E.; Currell, K.; Clarke, J.; Cole, J.; Blannin, A.K. Effect of beverage glucose and sodium content on fluid delivery. Nutrition & Metabolism 2009, 6, 9, doi:10.1186/1743-7075-6-9.
Reviewer 2 Report
Comments and Suggestions for Authors
Thank you for allowing me to review this manuscript. Excellent work from the authors, a really well written manuscript. Yet, allow me to mention some points that I believe will improve the manuscript.
From the beginning until line 65:
Well written, although the statements are too strong, especially for cognitive function where literature findings remain largely equivocal. On a personal note I would agree with the authors, however since we are talking about a review paper, caution should be taken. Please add more information on possible cognitive function and hypohydration if the authors wish to have a stronger and more substantiated argument on this matter.
L64 hypohydration induced strain also leads to increased epinephrine, increased/early glycogen breakdown and acceleration of fatigue mechanisms, if the authors wish to include these in the manuscript.
L109 BHI measurement may impacted by age. How does age impact this? Does BHI increase of decrease with age, and which age range?
Section 3.1 Electrolytes
L172 “longer duration and higher intensity exercise”. This description is too vague the way it is mentioned in the manuscript. Please define “long duration”, either in terms of minutes, hours, or give more specific examples (i.e. marathon, football match). Similarly, please define “high intensity”.
Section 3.2 Carbohydrates
L235-236 “fructose-to-glucose ratios exceeding one are typically discouraged in CES as they in-235 crease lumen osmolality due to limited fructose absorption”. It would be nice to have a reference for this.
Section 3.4 Other potential ingredients
L 346. I would like to report that on milk rehydration trials, lactose intolerant individuals can suffer quite a lot because of diarrhea, rendering any water “retention” pretty much questionable. Energy drained (due to diarrhea) individuals would also have significantly decreased athletic performance. I don’t know whether the authors would like to add this information here.
L451-452:” glycerol is an osmotically active solute that is rapidly absorbed after oral intake and 451 evenly distributes to all body water compartments”. I am not sure that glycerol enters the muscle quickly enough. Glycerol can take up to 3 hours to enter muscle cells. When muscles are exposed to a hyperosmotic environment (as when drinking a glycerol solution) initially lose water and muscle cells shrink. Only later, as glycerol enters the intracellular space the muscles start to restore their volume. Therefore, water retention and plasma volume increases could also be partly attributed to water coming out of the muscles. Plasma volume increases, muscle hydration decreases, something that could have a totally different outcome on muscle function and exercise performance.
L517, 597: “High intensity” and “long duration”. Please be more specific. For example, in L512-513 where intensity and duration are perfectly described.
L597 there is a conclusion with regards to high intensity exercise, where all the reported references’ intensity did not go beyond 65% VO2max. Cycling and even marathons are contested at a much higher intensity. It would be nice if some practical or real-world scenario situations were included in this section
Author Response
Reviewer 2
Comments and Suggestions for Authors
- Thank you for allowing me to review this manuscript. Excellent work from the authors, a really well written manuscript. Yet, allow me to mention some points that I believe will improve the manuscript.
We want to express our gratitude to the reviewer for the thoughtful review, and the feedback received. We hope to have address all points mentioned with the purpose of improving our manuscript.
- From the beginning until line 65:
Well written, although the statements are too strong, especially for cognitive function where literature findings remain largely equivocal. On a personal note I would agree with the authors, however since we are talking about a review paper, caution should be taken. Please add more information on possible cognitive function and hypohydration if the authors wish to have a stronger and more substantiated argument on this matter.
We agree that, while the general believe is that hypohydration might impair several outcomes of performance as stated in the section, some analyses of the evidence have suggested otherwise, particularly regarding cognitive performance (i.e., [1]). We thank the reviewer for pointing this out, and we soften the statements made in the section (lines 42;57;58;62) (highlighted) (references provided at the end of this document).
- L64 hypohydration induced strain also leads to increased epinephrine, increased/early glycogen breakdown and acceleration of fatigue mechanisms, if the authors wish to include these in the manuscript.
We deeply appreciate the input received. We added these aspects to lines 67-68 (highlighted) and support the statements with references.
- L109 BHI measurement may impacted by age. How does age impact this? Does BHI increase of decrease with age, and which age range?
We thank the reviewer for pointing this out. The citation was incorrect, and we replaced it with a correct one, specifying how age might impact BHI based on the study results (line 113-115) (highlighted). We also checked all other references throughout the manuscript.
- Section 3.1 Electrolytes
L172 “longer duration and higher intensity exercise”. This description is too vague the way it is mentioned in the manuscript. Please define “long duration”, either in terms of minutes, hours, or give more specific examples (i.e. marathon, football match). Similarly, please define “high intensity”.
Cited studies reported increased sodium sweat losses with longer exercise duration and higher intensity of exercise, rather than specific settings. We changed the sentence accordingly to improve clarity (line 184) (highlighted).
- Section 3.2 Carbohydrates
L235-236 “fructose-to-glucose ratios exceeding one are typically discouraged in CES as they in-235 crease lumen osmolality due to limited fructose absorption”. It would be nice to have a reference for this.
This was also noted by another reviewer. We added a citation and further elaborated on the subject (line 251-253) (highlighted).
- Section 3.4 Other potential ingredients
L 346. I would like to report that on milk rehydration trials, lactose intolerant individuals can suffer quite a lot because of diarrhea, rendering any water “retention” pretty much questionable. Energy drained (due to diarrhea) individuals would also have significantly decreased athletic performance. I don’t know whether the authors would like to add this information here.
We agree with reviewer’s view of the challenges related to milk as a rehydration beverage and stated these aspects in lines 383-384 (highlighted). A new citation was added to support the statement.
- L451-452:” glycerol is an osmotically active solute that is rapidly absorbed after oral intake and 451 evenly distributes to all body water compartments”. I am not sure that glycerol enters the muscle quickly enough. Glycerol can take up to 3 hours to enter muscle cells. When muscles are exposed to a hyperosmotic environment (as when drinking a glycerol solution) initially lose water and muscle cells shrink. Only later, as glycerol enters the intracellular space the muscles start to restore their volume. Therefore, water retention and plasma volume increases could also be partly attributed to water coming out of the muscles. Plasma volume increases, muscle hydration decreases, something that could have a totally different outcome on muscle function and exercise performance.
We appreciate this very interesting comment. In fact, we are aware of classic research suggesting that marked increases in plasma osmolality due to glycerol intake might produce muscle tissue dehydration [2]. In the same vein, other authors have suggested that restoration of plasma volume with glycerol intake occurs before than that of interstitial and intracellular fluid compartments [3]. We conclude that our statement was not substantiated and changed the sentence accordingly (line 502) (references provided at the end of this document).
- L517, 597: “High intensity” and “long duration”. Please be more specific. For example, in L512-513 where intensity and duration are perfectly described.
We specified “high intensity “and “long duration” as per cited in the consensus document (line 567) (highlighted).
- L597 there is a conclusion with regards to high intensity exercise, where all the reported references’ intensity did not go beyond 65% VO2 Cycling and even marathons are contested at a much higher intensity. It would be nice if some practical or real-world scenario situations were included in this section.
We thank the reviewer for pointing this out. In fact, none of the cited studies evaluating beverages containing different sodium concentrations during exercise reported exercise intensities >70% VO2max. We checked a systematic review (2018) evaluating the impact of sodium ingestion during exercise on endurance performance [4], but we could not find additional studies exploring high-intensity exercise protocols. We, thus, removed the comment about intensity and duration (line 655), and added this as a potential research gap in Table 3 (in line with another reviewer) (lines 824-825) (highlighted) (references provided at the end of this document).
References
- Goodman, S.P.J.; Moreland, A.T.; Marino, F.E. The effect of active hypohydration on cognitive function: A systematic review and meta-analysis. Physiology & Behavior 2019, 204, 297-308, doi:https://doi.org/10.1016/j.physbeh.2019.03.008.
- Gleeson, M.; Maughan, R.J.; Greenhaff, P.L. Comparison of the effects of pre-exercise feeding of glucose, glycerol and placebo on endurance and fuel homeostasis in man. Eur J Appl Physiol Occup Physiol 1986, 55, 645-653, doi:10.1007/bf00423211.
- van Rosendal, S.P.; Osborne, M.A.; Fassett, R.G.; Coombes, J.S. Guidelines for Glycerol Use in Hyperhydration and Rehydration Associated with Exercise. Sports Medicine 2010, 40, 113-139, doi:10.2165/11530760-000000000-00000.
- McCubbin, A.; Costa, R. Impact of Sodium Ingestion During Exercise on Endurance Performance: A Systematic Review. International Journal of Sports Science 2018, 8, doi:10.5923/j.sports.20180803.05.